# What distinguishes 100-year precipitation extremes over Central European river catchments from more moderate extreme events?

Florian Ruff and Stephan Pfahl

Freie Universität Berlin, Institute of Meteorology, Carl-Heinrich-Becker-Weg 6-10, 12165 Berlin, Germany

**Correspondence:** Florian Ruff (florian.ruff@met.fu-berlin.de)

**Abstract.** Historical extreme flooding events in Central European river catchments caused high socioeconomic impacts. Previous studies analysed single events in detail but did not focus on a robust analysis of the underlying extreme precipitation events in general as historical events are too rare for a robust assessment of their generic dynamical causes. This study tries to fill this gap by analysing a set of realistic daily 100-year large-scale precipitation events over five major European river catchments with the help of operational ensemble prediction data from the ECMWF. The dynamical conditions during such extreme events are investigated and compared to those of more moderate extreme events (20- to 50-year). 100-year precipitation events are generally associated with an upper-level cut-off low over Central Europe in combination with a surface cyclone southeast of the specific river catchment. The 24 hours before the event are decisive for the exact location of this surface cyclone, depending on the location and velocity of the upper-level low over Western Europe. The difference between 100-year and more moderate extreme events vary from catchment to catchment. Dynamical mechanisms such as an intensified upper-level cut-off low and surface cyclone are the main drivers distinguishing 100-year events in the Oder and Danube catchments, whereas thermodynamic mechanisms such as a higher moisture supply in the lower troposphere east of the specific river catchment are more relevant in the Elbe and Rhine catchments. For the Weser/Ems catchment, differences appear in both dynamical and thermodynamic mechanisms.

## 1 Introduction

Flooding events in highly populated areas are one of the main natural hazards for the modern society (Hammond et al., 2015; Kreibich et al., 2014; Barredo, 2007). Central Europe was regularly affected by substantial flooding events in the last decades in several river catchments due to extreme precipitation events. Remarkable flooding events were the Rhine floods in June 2021 and in December 1993, the Oder floods in May 2010 and in July/August 1997 as well as the flooding events in Central Europe in June 2013 and in August 2002. During the two latter events, several catchments, but mainly the Danube and Elbe, were affected. Such an impact on several river catchments often occurs with spatially very extended extreme precipitation events. Many of the flood events had devastating impacts such as several to hundreds of human life losses, thousands of evacuations as well as economic losses of several million to billion euros (Munich Re, 2022; Engel, 2004; Munich Re, 1999). The Central European flood in August 2002 was the most expensive natural hazard for Central Europe, caused accumulated costs of about 15 billion euros for both public and private properties and even impacted business interruptions after the event (Mueller,

2003), while the Rhine flood in June 2021 was globally the second most expensive natural hazard in 2021 (Statista, 2022). These damages were associated with high water levels, partly even the highest water levels on record. For instance, during the Central European flood in June 2013, several gauges measured record water levels including the German city Passau (at the Danube) where the highest water levels of about 12.75 m were recorded since 1501 (Merz et al., 2014). However, several

improvements in flood risk management implemented in the beginning of the 21th century following the floods at the end of the 20th century prevented some areas from socioeconomic losses to extend that occurred earlier (Merz et al., 2014; Bissolli et al., 2011). Nevertheless, as losses can never be totally eliminated, it is important to gain more knowledge about the atmospheric triggering mechanisms that induce such extreme floods to improve forecasts of these events and warn the public at an early stage. Additionally, flooding events can become even more relevant in the future as climate simulations show a higher frequency

of extreme precipitation events in the mid-latitudes in a warmer climate (Pendergrass and Hartmann, 2014; Fischer et al., 2013; O'Gorman and Schneider, 2009).

Central European flooding is typically associated with large-scale extreme precipitation events over a river catchment, which are linked to mid-latitude cyclones that become quasi-stationary over Central or Eastern Europe for a few days as described in Grams et al. (2014); Pfahl (2014); Blöschl et al. (2013); Bissolli et al. (2011); Ulbrich et al. (2003). In case studies, Grams

et al. (2014) and Ulbrich et al. (2003) found that the origin of such a circulation anomaly was linked to upper-level Rossby wave breaking and the formation of an upper-level cut-off low over Western Europe. The stationarity of such a cut-off low can be facilitated by a blocking anticyclone over the North Atlantic, sometimes in combination with a blocking over Western Russia as shown by Blöschl et al. (2013). With such an upper-tropospheric configuration, either a surface cyclone moving from Western Europe southeastwards around the southern flanks of the Alps, or cyclogenesis over the Ligurian Sea, followed by a

northeastward track of the cyclone towards Central or Eastern Europe, is often associated (Bissolli et al., 2011; Mudelsee et al., 2004; Ulbrich et al., 2003). Such cyclone tracks are typically known as a "Vb" weather type (van Bebber, 1891). However, the Central European flood in June 2013 was triggered by surface cyclones that formed over Eastern Europe but moved towards Central Europe as well (Grams et al., 2014). Although a surface cyclone is often of high relevance for producing extreme precipitation, it is not necessary that this cyclone is associated with extremely low pressure (Pfahl and Wernli, 2012). This was

also shown by Grams et al. (2014), as the surface cyclones during the Central European flood in June 2013 were rather shallow with minimum pressure between 995 hPa and 1000 hPa. Hence, other important processes such as the transport of atmospheric moisture and the quasi-stationarity of the weather pattern have to be taken into account.

Surface cyclones can facilitate the transport of atmospheric moisture in the lower troposphere towards a certain area. For extreme flooding events, typically several moisture sources are important. On the one hand, moisture can evaporate into the

atmosphere from the North Atlantic as well as from the Mediterranean (Blöschl et al., 2013; Sodemann et al., 2009; Ulbrich et al., 2003). On the other hand, a high amount of moisture evaporates over continental areas such as Central and Eastern Europe along the tracks of cyclones (Grams et al., 2014; Winschall et al., 2014). Especially during the northward movement of "Vb" cyclones towards Central or Central Eastern Europe, continental moisture sources become more dominant (Winschall et al., 2014; James et al., 2004). Regional ascent of these moist air masses occurs shortly before and during the extreme precipitation

events over the specific river catchment. The orographic effects of the Alps and other mountainous areas additionally play an

important role for the location of the extreme precipitation as it was the case in several of the extreme flooding events (Grams et al., 2014; Szalińska et al., 2014; Blöschl et al., 2013; Ulbrich et al., 2003).

However, just the occurrence of an extreme precipitation event does not necessarily trigger a major flood event. Besides the precipitation intensity, the time period over which precipitation accumulates is an important factor. While short-term extreme precipitation often triggers flash floods, precipitation accumulated over at least several hours up to several days is more likely to cause large flooding events. As the case studies cited above show, also hydrological preconditions such as an (almost) saturated soil and a high river runoff already prior to the event are important prerequisites for the development of a flooding as well. Such conditions are often the result of the accumulation of precipitation during the days or weeks before the extreme flooding (Grams et al., 2014; Merz et al., 2014; Bissolli et al., 2011; Engel, 1997).

In most previous studies, detailed analyses of the atmospheric dynamics of historical extreme precipitation events associated with Central European flooding were done in case studies. Some of these studies compared their results with a few other floods in the same river catchment. However, due to the shortage of the observational record it is difficult to determine the return level of the most severe historical floods, and there are too few events to systematically investigate the generic atmospheric processes leading to such events. Today's knowledge about the mechanisms of such extreme events is, therefore, based on analyses of single events as well as on statistical investigations of more moderate precipitation events with return periods on the order of one up to a few years (Donat et al., 2013; Sillmann et al., 2013; Pfahl and Wernli, 2012; Kenyon and Hegerl, 2010). Higher return levels are typically estimated based on extreme value theory such as in Maraun et al. (2011). Still, the length of observational time series plays an important role for the precision of such statistical estimates as well. Additionally, a detailed analysis of the underlying meteorological processes is complicated as the extreme events are usually not explicitly identified in such statistical assessments. Kelder et al. (2020) recently showed that a large ensemble simulation approach can yield novel insights into more extreme events and their changes during the last decades. They investigated 3-day precipitation extremes with return periods of up to 100 years in Norway and showed that a 100-year precipitation event in 1981 is equivalent to a 40-year event in 2015. Nevertheless, there is still a systematic lack of knowledge on the mechanisms leading to very extreme precipitation events with return periods on the order of 100 years. It is also unclear how these mechanisms of 100-year events distinguish from more moderate precipitation events. These knowledge gaps are addressed in the present study.

This study focuses on the dynamical mechanisms behind very extreme large-scale precipitation events with a return period of 100 years in five major Central European river catchments, which have the potential to cause devastating flooding events. In order to robustly analyse these mechanisms, we use data from operational ensemble weather prediction simulations to generate a large set of multiple realisations of possible weather situations for a quasi-stationary climate following Breivik et al. (2013). This set of multiple possible realisations has an equivalent length of 1200 years and, hence, is several times longer than conventional observational time series. Several extreme precipitation events with return periods of at least 100 years are identified in this dataset with the help of extreme value theory. Estimates of such extreme return levels from observational time series are compared with the ensemble prediction data to evaluate the realistic representation of extreme precipitation

magnitudes. The atmospheric processes associated with the 100-year events are then systematically analysed. In a second step, the results are compared to the conditions during more moderate extreme precipitation events.

The following section 2 describes the ensemble prediction dataset that is used for the identification and meteorological analysis of 100-year precipitation events. In addition, three observational datasets are introduced that are used for verification. All methodological aspects regarding the processing of ensemble data and further statistical methods are explained in section 3. In section 4, both the atmospheric processes associated with 100-year events and the differences to less extreme precipitation events as well as an analysis of tracks, frequencies and intensities of objectively identified surface cyclones during the extreme events are presented. Finally, conclusions and a discussion of our most important findings are provided in section 5.

## 2 Data

For this study, a large dataset of daily precipitation events is constructed from operational ensemble weather prediction data. Additional observational data are used to evaluate this model-based dataset. All these datasets are described in the following.

### 2.1 Ensemble prediction data

The archive of the operational weather prediction model of the Integrated Forecasting System (IFS), provided by the European Center for Medium-Range Weather Forecasts (ECMWF), is used to get a large set of simulated but realistic daily weather and precipitation events from their ensemble prediction system (EPS). The workflow of the EPS is described in more detail in Molteni et al. (1996). The IFS is a comprehensive earth system model, combining the atmospheric model of ECMWF with community models for other components of the earth system and a data assimilation system (ECMWF, 2023c). It is used for all forecasting activities of ECMWF. The full documentation of the model and the assimilation system can be found in ECMWF (2023b). Using an operational weather model ensemble instead of a single climate model initial-condition large ensemble (SMILE) has the advantage that the data are available with a higher spatial resolution and that the weather model is very well calibrated due to extensive comparison with observations on a daily basis, even though surface precipitation observations are not assimilated. On the contrary, the weather prediction ensemble may suffer from temporal inhomogeneities and inter-dependence between ensemble members, which is not the case for SMILEs. Although SMILEs can be used in a similar manner to obtain large sample sizes, the approach of using weather prediction ensemble data for extreme event analyses is still understudied. The inter-dependence between ensemble members is investigated in section 3.2 while temporal inhomogeneities as well as additional limitations of our approach are discussed in more detail in section 5.

On every day, ensemble simulations are started with lead times of at least 10 days and 51 ensemble members. One member represents the control run with no perturbed initial conditions while the other 50 member represent perturbed runs with slightly changed initial conditions between each ensemble member and additional stochastic perturbations introduced during the model integration. Since the 25th of March 2003, these ensemble simulations are operated two times a day, starting at 0 and 12 UTC. The combination of 51 ensemble members and two daily initialisations results in 102 simulations per real day. The basis for

this study are daily precipitation sums from these forecasts computed by adding up the large-scale and convective precipitation over 24 hour time periods contiguously.

Instead of the daily precipitation sums for all 10 days of each simulation, just the 10th day of each simulation (accumulated precipitation between forecast hour 216 and forecast hour 240, resulting in a 24-hour time period each) is used in this study, following Breivik et al. (2013). They used a large dataset from ensemble simulations to estimate return values of wave heights over the oceans. To this end, Breivik et al. (2013) assumed that the weather conditions, and especially the wave heights, of the 10th day of each simulation are independent between the different ensemble members due to the advanced lead time. During the first days, the simulations of the different member are correlated due to a strong dependency on the initial conditions, but this correlation weakens when the forecasts advance in time. Breivik et al. (2013) also performed several statistical evaluations to demonstrate this independence on the 10th forecast day and to evaluate the statistics of the simulated wave heights in comparison to observations. Also for the daily precipitation events investigated in this study, due to the high spatial and temporal variability of precipitation, we assume that the weather conditions of each 10th simulation day do not strongly correlate with the initial conditions of the simulations and the daily precipitation events can thus be regarded as independent realisations. Following Breivik et al. (2013), this assumptions is evaluated statistically as described in section 3.2.

There were several IFS model updates between the years 2003 and 2019 that may influence the modelled precipitation amounts as well as the forecast skill at advanced lead times and, thus, the results of our analyses. A collection of all changes (from Cycle 25r1, implemented before 1st January 2003, to Cycle 46r1, last implemented cycle before 31th December 2019) can be found in ECMWF (2023a) while the full documentation (incl. data assimilation, dynamical processes and parameterisation) of all the individual IFS model cycles is provided in ECMWF (2023b). Typically, several small changes/improvements were implemented in each new model cycle, and precipitation forecasts were slightly improved over several years, but no notable large improvement can be identified for a specific model cycle. Noteworthy updates have been a new formulation of the humidity analysis (Cycle 26r1), a new moist boundary layer scheme (Cycle 29r1), improved precipitation forecasts over Europe due to several technical changes such as bias corrections and assimilation improvements (Cycle 32r3), changes in the cloud scheme (Cycle 41r1) and improvements in near-coastal precipitation forecasts due to changes in cloud physics (Cycle 45r1) as well as several changes in the data assimilation and other technical improvements. In order to evaluate the effect of these updates on the dataset, the temporal distribution of extreme precipitation events, including possible trends, over the years is studied in section 3.3.

First tests showed that very high percentiles (99th, 99.9th and 99.99th), which represent the most extreme precipitation events of this daily dataset, are systematically higher for the first years of the ensemble prediction data (see Supplementary Fig. S1 for the Danube catchment). Since 2008, the data show a more consistent amplitude of these different percentiles, comparable to the observational datasets (not shown). Hence, to exclude any possible inconsistencies due to temporal inhomogeneity of the data, just the ensemble prediction data from the 1st of January 2008 until the 31th December 2019 are used in the following. No trend can be detected in the percentiles shown in Supplementary Fig. S1 during this restricted period according to the Mann-Kendall test (95% confidence level), except for the 99.9th percentile in the Weser/Ems and Elbe catchments. The combination of 102

different simulated precipitation events per real day and 12 years of simulations from the archive of the EPS leads to an overall
dataset of 1224 years of simulated but realistic daily precipitation events. The data cover the entire globe and are available on
a regular lon/lat grid, whose resolution changes over time due to updates of the model system. Since these precipitation data
are further used to compute daily precipitation sums for large Central European river catchments (see section 3), they are first
interpolated on a relatively coarse grid of $1° \times 1°$.

To study the atmospheric conditions during and before the most extreme daily precipitation events, several additional mete-
orological parameters are extracted from simulations that included an extreme precipitation event (as defined below) and used
for composite and single event analyses during and up to four days ahead of the events. All parameters are available every six
hours from forecast hour 120 (four days before the event occurred) until forecast hour 240 (the end of the event) on a regular
lon/lat grid with a spatial resolution of $0.1° \times 0.1°$. These data cover a large European region, ranging from $30°W$ to $45°E$ and
from $25°N$ to $75°N$.

## 2.2 Observational datasets

Three observational precipitation datasets (REGEN, HYRAS and E-OBS) are used to evaluate the precipitation climatologies
obtained from the ensemble simulations. These datasets are all based on station observations and mainly differ in their spatial
resolution, region of coverage and method used for the interpolation of available station data to a regular spatial grid.

### 2.2.1 REGEN data

The Rainfall Estimates on a Gridded Network (REGEN) dataset (Contractor et al., 2020) is a observational precipitation
dataset covering global land areas. It is based on the spatial interpolation of in situ surface precipitation measurements from
several large observational archives, such as the Global Historical Climate Network Daily, hosted by the National Oceanic
and Atmospheric Administration, and the Global Precipitation Climatology Centre, hosted by Deutscher Wetterdienst. All
observed precipitation time series are quality controlled before interpolation. The station density differs strongly between
specific regions and continents. Africa and Central Asia have a low density of stations while many parts of North America,
Australia and especially Europe have a high density. Besides daily precipitation sums, this dataset provides parameters like the
standard deviation of these precipitation sums as well as the number of rain gauges within each grid box. For this study, just
the daily precipitation sums of Version 1-2019 based on around 135.000 stations are used for further investigations. These data
are available on a regular lon/lat grid for all global land areas with a spatial resolution of $1° \times 1°$ from the 1st of January 1950
until the 31th of December 2016.

### 2.2.2 HYRAS data

The HYRAS dataset represents a climatology of daily precipitation sums for Germany and river catchments of major rivers in
neighbouring countries, provided by Deutscher Wetterdienst (DWD). This dataset is based on thousands of in situ observations.
The number of stations increases over time, and the station density differs between countries and is much higher in Germany

and the Netherlands compared to parts of, e.g., Poland and France. All these observational station data were quality controlled by the DWD and were interpolated to a Lambert Conformal Conic Projection by using the REGNIE method, which is described in more detail in Rauthe et al. (2013). Here, these data are interpolated again on a regular lon/lat grid with a spatial resolution of 0.05°x 0.05°. They are available from the 1st of January 1951 until the 31th December 2015 and cover an area of Germany and adjacent river catchments ranging from about 1.85°E to 20.80°E and 45.10°N to 55.65°N.

### 2.2.3 E-OBS data

The E-OBS dataset is an on-going observational dataset for European land areas, provided by the European Climate Assessment & Dataset (ECAD) initiative. This dataset combines observational data from a large set of stations across several countries, provided by National Meteorological Services, in an ensemble approach with 100 ensemble members for several meteorological parameters such as temperature and precipitation, as described in Cornes et al. (2018). The observational data are quality controlled, both by each National Meteorological Service and by the ECAD itself. As for the other datasets, the temporal and especially the spatial availability of station data differs, especially across national borders, with higher densities in Central and Northern Europe. Through the ensemble simulations, the uncertainties of the interpolation of the data to a regular grid is reduced. For this study, just the daily precipitation sums are used. Based on E-OBS version 21.0e, these precipitation data are available on a regular lon/lat grid with a spatial resolution of 0.1°x 0.1° from the 1st of January 1950 until the 31th of December 2019 and cover a large European region, ranging from 25.0°W to 45.5°E and 25.0°N to 71.5°N.

## 3 Methodology

In this section, the methodology to identify and analyse extreme precipitation events in ensemble prediction data is described, starting with the catchment definitions of the selected rivers, followed by the statistical evaluation of the suitability of the ensemble data. Afterwards, the determination of the extreme events is introduced as well as the composite analysis and cyclone identification and tracking method that are used for characterising the underlying atmospheric processes.

For simplicity, all figures and analyses in section 3.2 are only shown for the Danube river catchment as representative for all catchments. In case that results for other catchments differ substantially from those of the Danube catchment, this is mentioned in the text.

### 3.1 River catchments

Five major Central European river catchments are considered in this study. Four river catchments represent the rivers Rhine, Elbe, Oder and the upper (most western) part of the Danube while the fifth catchment covers the region of the two rivers Weser and Ems in Northern Germany, here denoted as Weser/Ems. As explained in section 1, flooding events in the past have shown that all these regions are affected by risks of flooding and associated socio-economic impacts during extreme precipitation events.

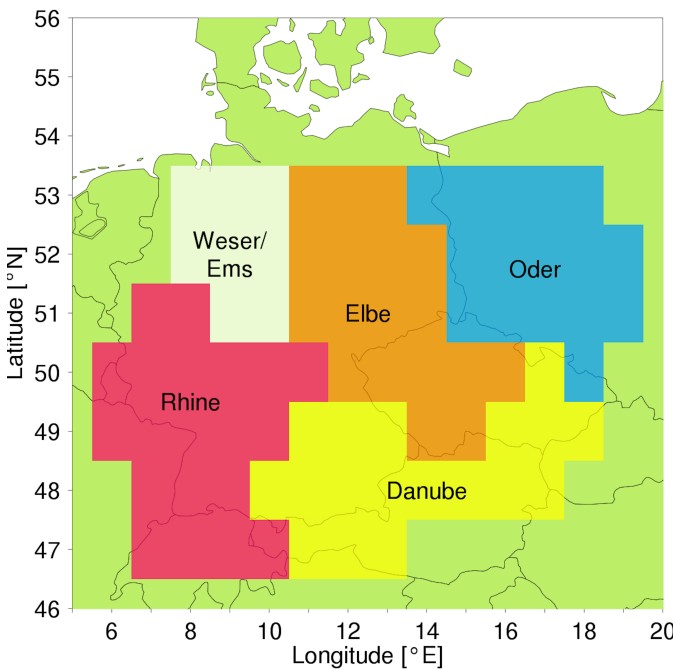

**Figure 1.** River catchments of the rivers Rhine, Elbe, Oder, Danube and the combination of the rivers Weser and Ems, denoted Weser/Ems, based on a 1°x 1° lon/lat grid over Central Europe.

The spatial coverage of these selected river catchments as adopted in this study is shown in Fig. 1. The spatial resolution of 1°x 1° is due to the resolution of the daily precipitation data of the ensemble prediction data (see section 2.1). Each grid point has been allocated to one of the five river catchments based on a river basin map by the Waterways and Shipping Administration of Germany (Wasserstraßen- und Schifffahrtsverwaltung des Bundes, 2022). Due to the relatively coarse resolution, this does not exactly reproduce the river catchments, but still provides a clear separation of the larger-scale precipitation patterns potentially associated with flooding in the different catchments.

Precipitation time series for each river catchment are constructed by averaging the time series from the associated 1°x 1° grid boxes, both from the ensemble predictions and observational datasets. Only for the comparison with HYRAS data, the methodology is slightly adapted because HYRAS does not cover the entire catchments shown in Fig. 1. Therefore, only 1°x 1° boxes with a HYRAS data coverage of at least 90% are considered for the comparison of simulated precipitation with the HYRAS observations.

## 3.2 Statistical evaluation of the ensemble prediction data

In order to evaluate the suitability of the precipitation dataset obtained from the ensemble weather prediction database for the analysis of extreme precipitation events, several statistical properties of the data are considered following Breivik et al. (2013). The following statistical evaluations cover the independence of events between ensemble members as well as within

a specific member and a comparison of the statistical distributions of daily precipitation between the members as well as with observational data. For these analyses, the 10th day (last 24 hours) of each ensemble forecast is taken into account, independent of the initialisation time. For each 24-hour period, the precipitation is spatially averaged over all grid points within the respective catchment to produce daily time series for each catchment and ensemble member. Figure 2 shows the results of the statistical evaluation exemplarily for the Danube catchment.

A basic assumption in the application of the ensemble prediction dataset as a climatological data basis for our analysis is that the precipitation accumulated over the 10th forecast day is independent between the ensemble members. To statistically evaluate this independence, Fig. 2a shows the statistical distribution of Spearman correlation coefficients between all possible combinations of ensemble members (5151 in total), in which low (high) correlations show a higher (lower) independence between ensemble members. The correlations are computed between time series as mentioned in the previous paragraph. The left box plot represents the correlations between the entire daily time series of two ensemble members with a total length of 12 years. These correlation coefficients vary between 0.12 to 0.32, with a mean correlation of about 0.19. All correlations are statistically significant, which is also due to the long time series. Physically such relative small correlations between the precipitation time series may arise from low frequency (longer than 10 days) variability in the climate system, which affects all ensemble members from the same initialisation date in a similar manner. However, in order to estimate 100-year return values and identify 100-year precipitation events, in this study we do not use the entire time series, but only the annual maxima of daily precipitation (block maximum approach, see section 3.3). Therefore, the right box plot in Fig. 2a indicates the distribution of correlation coefficients between such annual maximum values (12 values for each ensemble member). Most of the correlation coefficients are around 0 (mean correlation of 0.04), but there is a huge spread between -0.90 and +0.85. In order to estimate the statistical significance in such a situation of multiple correlation coefficients, we apply the False Discovery Rate (FDR) test of Benjamini and Hochberg (1995), as described in Ventura et al. (2004), to the entire sample of p-values associated with the individual correlation coefficients from the various combinations of ensemble members. According to this FDR test, only one out of the 5151 correlation coefficients for the Danube catchment can be considered as significant, and this number of significant correlations varies between zero and two for the other catchments. These very low numbers of significant correlation coefficients, together with the relatively small correlations of the entire time series, justify our assumption of independence, in particular when considering extreme (such as annual maximum) daily precipitation events.

A second important criterion for the applicability of the ensemble weather prediction data to analyse extreme precipitation events is a realistic representation of observed precipitation statistics. This is evaluated by comparing quantiles from the statistical distributions of daily precipitation between the ensemble data (taking all members together) and three observational datasets (REGEN, HYRAS and E-OBS) in quantile-quantile plots (QQ-plots), as shown in Fig. 2b. A perfect match of the distributions would be indicated by quantiles falling on the diagonal. This is the case for precipitation intensities below about 20 mm, which occur most frequently (see Fig. 2c). In this range, the distributions of daily precipitation from the ensemble data and the three observational datasets are thus almost equal. For higher intensities, there are some differences between the quantiles, with the ensemble data overestimating the higher quantiles compared to E-OBS data, but underestimating them in comparison to HYRAS. The deviations from the diagonal are mostly smaller in other catchment areas (see Supplementary Fig. S3), which

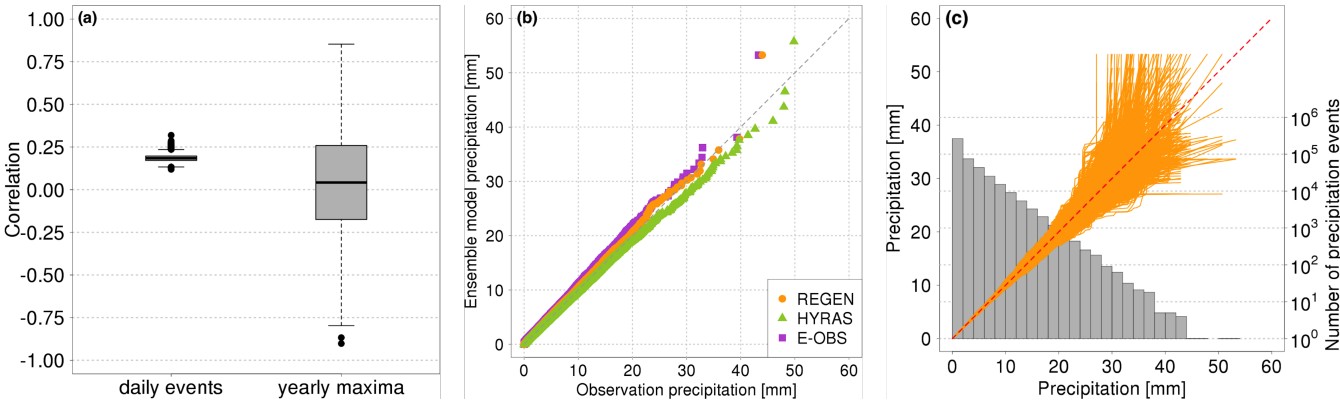

**Figure 2.** Statistical evaluation of daily precipitation data obtained from ensemble weather predictions for the Danube catchment. **(a)** Distribution of Spearman correlation coefficients between (left) daily precipitation time series and (right) annual maximum daily precipitation values for every combination of ensemble members (5151 in total). The box is bordered by the 25th and 75th percentile, while the whiskers extend to the most outer point which is not more than 1.5 times the interquartile range from the box. **(b)** Quantile-quantile-plot of daily precipitation from the ensemble weather prediction data (vertical axis) and three observational datasets (REGEN, HYRAS and E-OBS; horizontal axis). Differences in simulated precipitation for HYRAS observations are due to the reduced area with HYRAS coverage (only grid points with at least 90% HYRAS coverage are taken into account). **(c)** Quantile-quantile-plot of daily precipitation from every ensemble member combination (5151 in total). Grey bars indicate the intensity distribution from the entire ensemble prediction dataset.

may be related to the relatively complex topography of the Danube catchments and the limited spatial resolution of both model and observational data. All together, this analysis shows that the ensemble weather prediction dataset represents the statistical distribution of daily precipitation reasonably well and lies within the observational uncertainties (differences between the observational datasets) even for high quantiles corresponding to extreme precipitation events.

     Finally, the statistical distributions of daily precipitation are also compared between the individual ensemble members.
Figure 2c shows the QQ-plots for every possible combination of ensemble members (5151 in total), each one as an orange line. The grey bars show the precipitation distribution of all daily events from all ensemble members. All QQ-plots show nearly identical distributions for all members and low precipitation intensities (20 mm and lower). For higher precipitation intensities, the QQ-plots start to diverge. Such a divergence is expected since the time series are 12 years long and a daily precipitation event of 30 mm roughly corresponds to a 10-year event. Random fluctuations in the occurrence of such rare events can thus
lead to large differences in the higher quantiles between the members.

     In conclusion, the analyses in this section have shown that the ECMWF ensemble prediction system produces realistic statistical distributions of daily precipitation in the selected catchments (see Fig. 2b) throughout each ensemble member (see Fig. 2c) and that extreme precipitation events obtained from the ensemble prediction dataset can be considered as statistically independent as the correlations between precipitation time series from different ensemble members are small, and (almost) no

| Catchments | Return levels | Confidence intervals | |
|:---:|:---:|:---:|:---:|
| Rhine | 34.49 mm | 33.28 mm | 35.84 mm |
| Weser/Ems | 42.56 mm | 39.80 mm | 45.32 mm |
| Elbe | 36.62 mm | 34.78 mm | 38.75 mm |
| Oder | 38.58 mm | 36.87 mm | 40.50 mm |
| Danube | 40.21 mm | 38.65 mm | 41.92 mm |

**Table 1.** Return levels and confidence intervals for 100-year daily precipitation events in different river catchments, obtained from the procedure in section 3.3.

significant correlations are obtained between extreme precipitation events in different members (see Fig. 2a). The dataset is thus considered to be suitable for a systematic analysis of very extreme, 100-year precipitation events.

### 3.3 Determination of extreme precipitation events

Building on the statistical evaluation presented in section 3.2, we now pool the daily precipitation data from all ensemble members together. To determine return levels corresponding to 100-year (and also 20- and 50-year) return periods and select the corresponding precipitation events, the block maximum approach from extreme value statistics is applied (see Coles et al., 2001). A yearly period is selected to determine block maxima, resulting in 1224 block maxima in total. Such a yearly block size is large enough to fulfil the Fisher-Tippett theorem, such that a generalised extreme value distribution (GEV) can be fitted to the block maxima by the maximum likelihood approach. Three parameters, the location ($\mu$), scale ($\sigma$) and shape ($\xi$) parameter are estimated to obtain the best fit of the extreme value distribution as it can be seen in Supplementary Fig. S4 for the Danube catchment. From these three parameters, the return level $l$ can be computed by the following equation:

$$
\begin{aligned}
l &= \mu + \sigma \cdot \frac{(x^\xi - 1)}{\xi} \\
x &= \frac{-1}{log(1 - \frac{1}{p})}
\end{aligned}
\tag{1}
$$

where a specific return period in years is denoted by $p$ following Stephenson (2002). Confidence intervals of the return level are obtained from bootstrap resampling (see Coles et al., 2001). A new set of block maxima is drawn with replacement from the original set and the return level is again obtained from a GEV fit and Eq. (1). This procedure is repeated 1000 times, leading to 1000 different return levels from which the 0.025 and 0.975 quantiles are used as confidence intervals.

Table 1 shows the 100-year return levels and confidence intervals estimated for the different catchment following this procedure. For the explicit analysis of 100-year events, all daily precipitation events in a specific river catchment with a precipitation amount above the 100-year return level are selected. This results in 13 100-year events for the Danube catchment (Rhine:

13, Weser/Ems: 10, Elbe: 11, Oder: 13). For comparison, more moderate extreme precipitation events that fall in between the 20-year and 50-year return level are chosen (Danube: 41 events, Rhine: 42, Weser/Ems: 29, Elbe: 38, Oder: 36). For simplicity, the 100-year precipitation events are abbreviated as MEPEs (most extreme precipitation events) and the more moderate, 20-50-year events as LEPEs (less extreme precipitation events) in the following.

The reasoning behind using LEPEs with return periods between 20 and 50 years for comparison is that we would like to compare MEPEs to a clearly distinct distribution of less extreme events. Of course, there are also events with return periods between 50 and 100 years, that is, in between the two groups. Nevertheless, since MEPEs and LEPEs generally occur in similar synoptic-scale environments, as discussed in detail in section 4, we do not expect such intermediate events to behave entirely different.

To further demonstrate the temporal homogeneity of this set of extreme events, we investigate their temporal distribution over the period 2008-2019, which is shown in Supplementary Fig. S2. First, no significant trend (according to the Mann-Kendall test, 95% confidence level) exists in the number of MEPEs and LEPEs per year over the 12 years (taking data from all catchments together to obtain a sufficient sample size). Based on the Kolmogorov–Smirnov test, again with a 95% confidence level, the temporal distribution of both MEPEs and LEPEs (again for all catchments together) over the 12 years cannot be distinguished from a Poisson distribution, which is the case for independent events with a constant mean rate. Finally, in the individual catchments the occurrences of the MEPEs and LEPEs per year almost all lie within the expected 95% confidence interval from a Poisson distribution with a constant mean rate (for MEPEs there is only one outlier in the Rhine catchment and for LEPEs two outliers in the Weser/Ems and Rhine catchments, which is both below 2% given the total numbers of 60 MEPEs and 186 LEPEs). In summary, all these tests are consistent with the hypothesis that the extreme events are distributed randomly over the entire period.

### 3.4 Composite analysis

To analyse the generic meteorological conditions associated with extreme precipitation events, composites are constructed by averaging an atmospheric field from a specific lead time over all events from a given catchment and intensity (100-year or moderate, 20-50-year events). Such a composite analysis emphasises the structures that are common to all or most of the events. In addition, all 100-year events have been examined individually, and specifically noticeable anomalies are mentioned in the text. By averaging fields from a fixed lead time, the composites are representative of a specific time step relative to the occurrence of the daily extreme event, but not of a specific time of the day, as the forecasts have been initialised at both 0 and 12 UTC. Composites of meteorological parameters with a strong diurnal cycle thus have to be interpreted with care. Differences between composite patterns are tested for statistical significance at each grid point using a Student's $t$-test on a significance level of 0.05 (see Coles et al., 2001).

## 3.5 Cyclone identification and tracking

As precipitation extremes in Europe are known to be linked to the occurrence of mid-latitude cyclones on the synoptic scale (Pfahl, 2014), the composite analysis is complemented by an automated cyclone identification and tracking method following Wernli and Schwierz (2006) in a slightly updated version as documented in Sprenger et al. (2017). Cyclones are identified and tracked based on 6-hourly sea level pressure (SLP) fields from the ECMWF ensemble forecasts. Cyclone centers are determined as local minima of the SLP, and subsequent cyclone centers are connected to form a cyclone track if they occur within a search area determined by the previous cyclone track. This cyclone tracking algorithm has been included in several recent inter-comparison studies (e.g. Flaounas et al., 2023; Neu et al., 2013). It has also been widely used to study the relationship between cyclones and extreme precipitation (e.g. Pfahl, 2014; Pfahl and Wernli, 2012).

Here, this algorithm is used to find the surface cyclone that is associated with a specific extreme precipitation event. For each event, all cyclone tracks are considered as possible candidates that exist at least up to six hours after the begin of the 24-hour event. The closest cyclone located in a specific area around the river catchment during that time step (Danube: 10°E-25°E & 40°N-50°N, Rhine: 5°E-20°E & 42°N-52°N, Weser/Ems: 5°E-20°E & 45°N-55°N, Elbe: 10°E-25°E & 45°N-55°N, Oder: 15°E-30°E & 45°N-55°N) is then associated with the event. These areas are mainly located southeast of the river catchments, where our composite analysis indicates a SLP minimum (see below). There are a few cases when no cyclone is located in the predefined area, when the SLP field is rather flat and does not possess a well-defined minimum. These cases are excluded from the analysis of cyclone tracks. Additionally, the cyclone tracks associated with the extreme events do not have the same length, such that the number of considered cyclones differs between lead times.

## 4 Results

In the first part of this section, the temporal and spatial distributions of 100-year and more moderate extreme events are shown (section 4.1). Afterwards, the main results of this study are presented covering the atmospheric conditions associated with 100-year events and their differences to more moderate events (section 4.2). The results of the cyclone identification and tracking approach are then shown for both types of events in section 4.3. Several figures and analyses are only shown for the Danube catchment, but results for other catchments are also discussed, in particular if they are substantially different.

### 4.1 Spatial and temporal distribution

#### 4.1.1 100-year precipitation events

In this section, 100-year events are characterised in terms of their spatial precipitation patterns and time of occurrence. The MEPEs are typically associated with high precipitation amounts over the entire river catchment. In most catchments, the composite precipitation patterns have a maximum near the center and a relatively flat decrease of the average daily precipitation towards the boundaries, except for the Danube (see Fig. 3a) and partly the Rhine catchment. There, the highest precipitation rate are found along the northern flank of the Alps, indicating the importance of orographic precipitation enhancement in

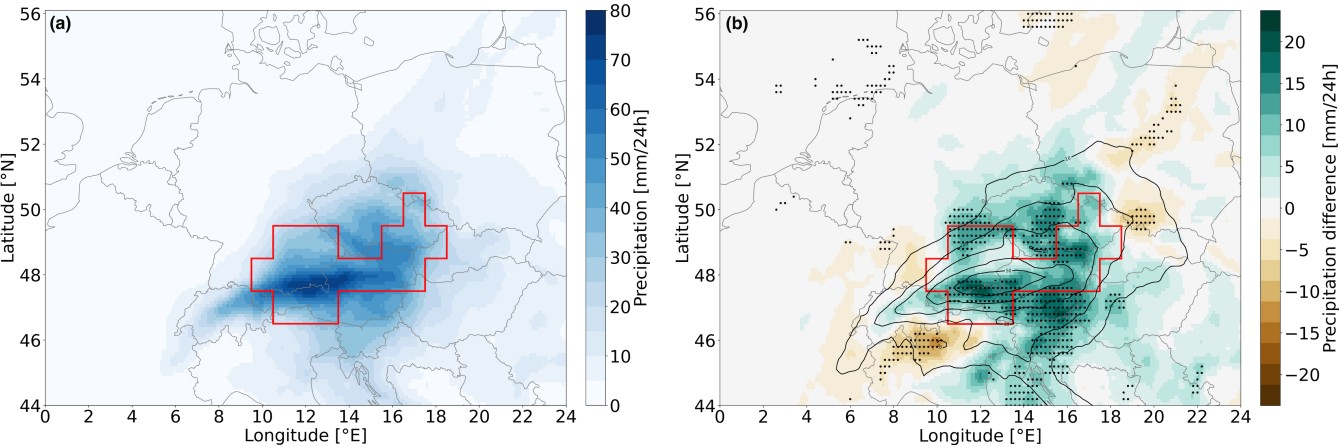

**Figure 3. (a)** Composite of the accumulated daily precipitation amount during all MEPEs. **(b)** Composite difference between the accumulated precipitation amount of MEPEs and LEPEs. Results are shown for events over the Danube catchment which is indicated by the red line. In **(b)**, positive (negative) values represent higher (lower) precipitation amounts during MEPEs, significant differences are marked by a black dot and the LEPE composite is shown as black contours (contour interval of 10 mm).

these two catchments. The mean composite precipitation amounts (spatially averaged over the catchment) are largest for the Weser/Ems and Danube catchments with 52.6 mm and 44.1 mm, respectively, followed by the Oder (42.1 mm), Elbe (40.9 mm) and Rhine (37.6 mm) catchments. The high values in the Weser/Ems region are associated with the relatively small catchment size. Higher values, of course, occur at single grid points. For the maximum grid-point values, the ranking changes as follows:
Elbe (77.7 mm), Danube (74.3 mm), Rhine (68.1 mm), Weser/Ems (63.8 mm) and Oder (62.7 mm).

Interesting aspects of 100-year precipitation events are their time of occurrence during the year and their temporal variability during the day of the event. The monthly frequencies of MEPEs for all river catchments are presented in Fig. 4a, which clearly shows that such events occur most frequently during the extended summer months June-September. While the frequency over, e.g., the Oder catchment is relatively high in July and August, the events over the Weser/Ems catchment are rather evenly
distributed over the warmer months. A few events, for instance over the Rhine catchment, also occur in May or October, which is, however, rather rare.

To quantify the temporal variability of precipitation during the day of the MEPEs, the six-hourly precipitation values of each event are ranked by their intensity, starting with the highest intensities, as shown in Supplementary Fig. S7 for the Danube catchment. Large differences between the red boxes would indicate that the daily MEPEs are mainly caused by very intense
six-hourly precipitation periods, whereas relatively equal distributions would indicate that the MEPEs are due to persistent precipitation during the entire day. For the Danube catchment, there is no clear indication that just one of these types dominates. There is some sub-daily variability during the MEPEs, with the six-hourly precipitation during the most intense period being about 75% higher than in the weakest period. Similar sub-daily distributions are found for the Rhine and Oder catchments. Just

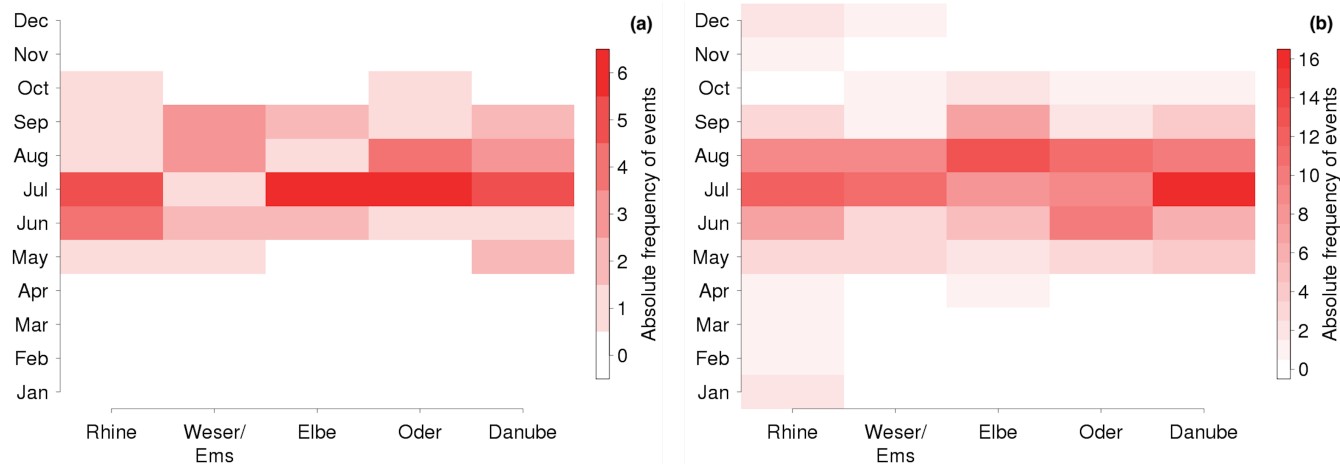

**Figure 4.** Monthly distribution of **(a)** MEPEs and **(b)** LEPEs for all five river catchments.

the Weser/Ems and Elbe events show a tendency towards larger differences, that is MEPEs being more influenced by intense
six-hourly precipitation.

### 4.1.2   Differences to 20-50-year precipitation events

By construction, the accumulated daily precipitation amounts for LEPEs are significantly smaller than for MEPEs, as shown
in Fig. 3b for the Danube catchment. Over the entire catchment, the MEPE composite indicates higher precipitation amounts
and most of these differences are statistically significant. The LEPE composite, which is displayed as black contours, shows
a similar spatial pattern as the MEPE composite with high precipitation amounts within the catchment and a maximum over
the northern part of the Alps. However, MEPEs are also associated with significantly higher precipitation rates over southeast
Germany, the Czech Republic as well as around the eastern parts of the Alps. In some regions in the western Alps and eastern
Europe, significantly more precipitation occurs during LEPEs. Also in the other river catchments, the spatial patterns of the
LEPE composites are similar to the corresponding MEPE composite (see Supplementary Fig. S5, mind the different colour
scale).

A comparison of the temporal occurrence and sub-daily variability of LEPEs illustrates a lot of similarities to the 100-year
events. The monthly frequencies of LEPEs (Fig. 4b) indicate a similar seasonality as for MEPEs. LEPEs occur most frequently
during the extended summer as well, mainly from May to October with a maximum frequency during June, July and August in
all river catchments. Hence, although the seasonality of LEPEs is less well-defined, there is no shift in the maximum frequency
towards other months compared to MEPEs. However, especially over the Rhine catchment, a few events also occur during
the other seasons. This low frequency of winter events is interesting given the fact that historical extreme floods in the Rhine
catchment occurred primarily in the winter season. Monthly distributions of even weaker extreme precipitation events in the
Rhine catchment, with return values of about 1-10 years, show rather equally distributed frequencies with slightly more events

in the winter season (see Supplementary Fig. S6 where HYRAS observations are used due to its high spatial resolution). The seasonal distribution of events in the Rhine catchment might thus shift from winter to summer with increasing event intensity.

Also the sub-daily variability of precipitation during LEPEs is similar to MEPEs (see again Supplementary Fig. S7). The precipitation differences between MEPEs and LEPEs are similar in all six-hourly sub-periods, indicating that the differences between the events cannot be explained by either differences in peak intensity or persistence alone. The same is true for the other river catchments, only the tendency towards larger differences between six-hourly intensities during MEPEs observed for the Weser/Ems and Elbe catchments is less pronounced for LEPEs.

## 4.2 Atmospheric conditions associated with extreme events

### 4.2.1 100-year precipitation events

In order to characterise the atmospheric conditions associated with 100-year precipitation events, composites of various fields 12 hours after the start of daily MEPEs over the Danube catchments are shown in Fig. 5. The temporal evolution of the geopotential height composites prior to the events is shown in Fig. 6. The geopotential height composite at 500 hPa (Fig. 5a) indicates a negative anomaly over Central Europe with its center southeast of the Danube catchment and the Alps, which has the form of an upper-level cut-off low. Similar cut-off low anomalies are found for the other river catchments, with the low pressure center typically located slightly east of the respective catchment (see Supplementary Fig. S8). The additional southward shift to the southern side of the Alps observed for the Danube catchment thus is slightly exceptional. In addition, positive geopotential height anomalies are located over the North Atlantic as well as over Eastern Europe (see again Fig. 5a). The presence of such ridges may lead to a quasi-stationary situation in which the cut-off low is prevented from moving farther east. In the sea level pressure composite for the Danube catchment (contours in Fig. 5a), an extended area of relatively low pressure covers large parts of Central and Eastern Europe, with the low pressure center being located close to and slightly east of the center of the upper-level cut-off low, which is a configuration conductive to baroclinic cyclone growth. An area of high pressure is found over the North Atlantic, underneath the 500 hPa ridge.

The upper-level cut-off low is also visible as positive potential vorticity (PV) anomaly on the 320 K isentrope (Fig. 5b). The center of the PV cut-off is shifted slightly southeastward compared to the center of the geopotential height anomaly at 500 hPa. Such stratospheric PV cut-offs or streamers are created by upper-level Rossby wave breaking (Grams et al., 2011) and have been shown to be often associated with extreme precipitation events in Europe (Martius et al., 2008). Also for the present case of MEPEs in the Danube catchment, the composite temporal evolution of the geopotential height field at 500 hPa (Fig. 6) indicates that the cut-off low develops from a trough located over northwestern Europe 24 hours before the event (Fig. 6a). Subsequently, Rossby wave breaking is likely associated with the separation of the negative geopotential height anomaly from the main northern region of low geopotential (high PV) and its southeastward displacement and slowdown south of the Danube catchment (Fig. 6b-d). The formation of the cut-off is associated with an intensification of the upstream and downstream ridges as well as surface cyclogenesis (see also section 4.3). MEPEs in the other catchments follow a similar development, with

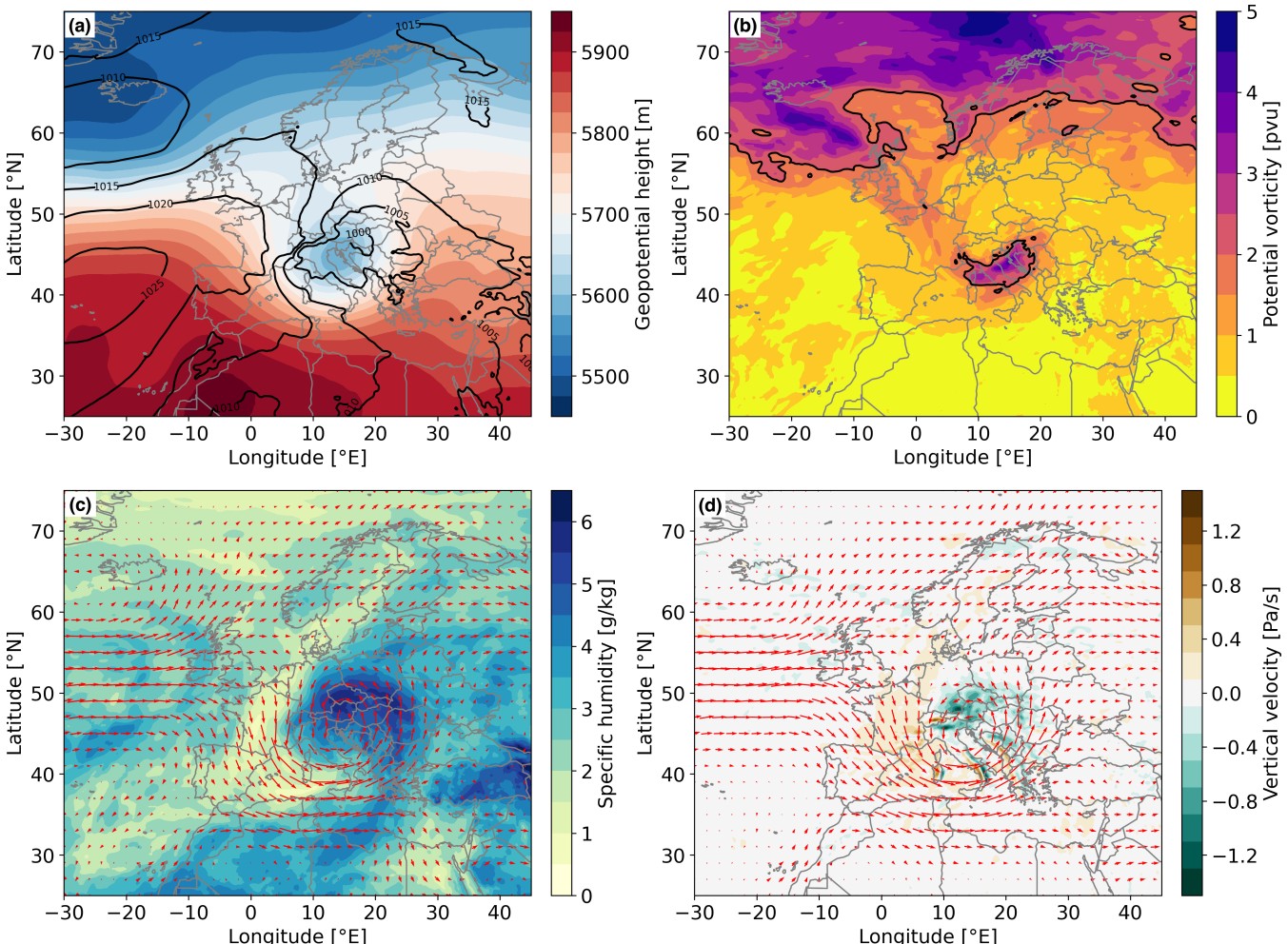

**Figure 5.** Composite of atmospheric conditions 12 hours after the start of daily MEPEs in the Danube catchment of **(a)** geopotential height at 500 hPa (colour shading) and sea level pressure (contours), **(b)** potential vorticity at 320 K, with the 2 pvu contour is indicated in black, **(c)** specific humidity (colour shading) and horizontal wind (red arrows) at 700 hPa as well as **(d)** vertical velocity (colour shading) and horizontal wind (red arrows) at 500 hPa.

the composite trajectory of the upper-level cut-off low and the location of surface cyclogenesis slightly displaced towards the respective catchment region (not shown).

The configuration of low pressure over Central Europe and high pressure over the North Atlantic and Eastern Europe is associated with northwesterly winds over Western Europe, westerly winds over the Mediterranean and southerly winds over Eastern Europe in the lower and middle troposphere (see Fig. 5c,d). Such a horizontal wind field facilitates the advection of moist air masses over the eastern Mediterranean and Eastern Europe towards Central Europe (Fig. 5c). In addition to the Mediterranean, continental evapotranspiration can act as an important (and often main) moisture source in such situations, as

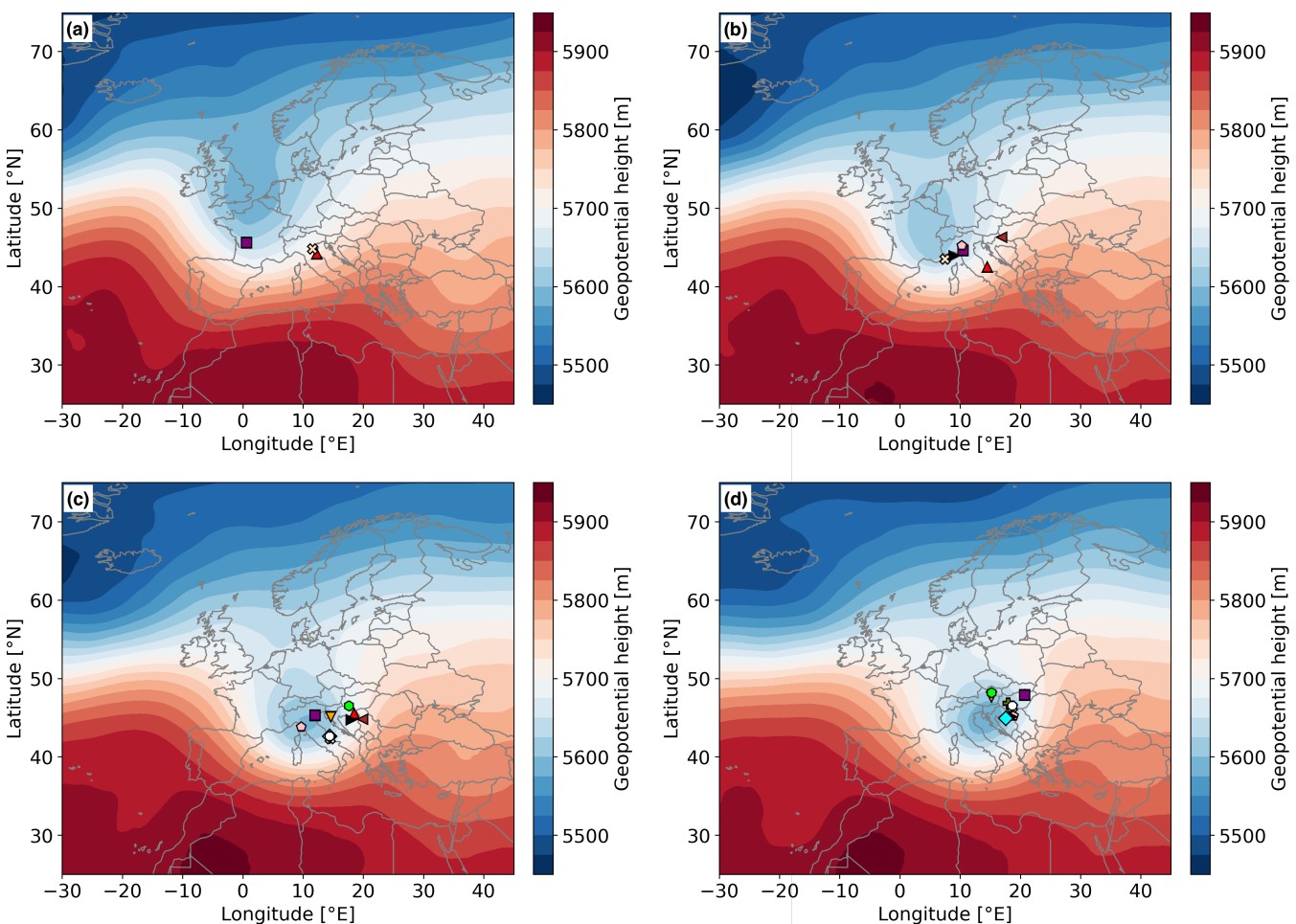

**Figure 6.** Composites of the geopotential height at 500 hPa for all MEPEs over the Danube catchment **(a)** 24 h before the event, **(b)** 12 h before the event, **(c)** at the beginning of the event as well as **(d)** 12 h after the beginning of the event. The surface cyclone centers (sea level pressure minima) from the cyclone identification and tracking approach are shown as coloured symbols. Since not every cyclone was identified for all time steps, later images show higher numbers of identified cyclone centers.

shown in previous case studies (Grams et al., 2014; Winschall et al., 2014; Sodemann et al., 2009; James et al., 2004). The moist air is transported towards the Danube catchment with northeasterly winds, where strong ascent (Fig. 5d) leads to rainout. Dynamical forcing, evident from positive vorticity advection, plays a role for this lifting in all catchments, but is weakest for the Danube (not shown). In addition, orographic effects can intensify the ascent, in particular in the case of the Danube (and also the Rhine) catchment, when the air masses are transported towards the Alpine ridge from the north. The slow displacement of the cut-off low can lead to continual ascent in the same catchment region and high accumulation of precipitation during the day of the MEPEs.

These composites show the average conditions during all 100-year precipitation events over a river catchment. Although most of the individual events develop in a similar way as shown in these composites, based on a visual analysis of the individual events, there are a few exceptions with a different progression. Two events over the Danube catchment are characterised by an upper-level low that moves north of the Alps, but a surface cyclone also developing east of the catchment (see Supplementary Fig. S9a for one case, 12 hours before the event). One event over the Weser/Ems catchment results from a small northward moving surface cyclone at the eastern flank of a large upper-level trough over the British Isles and the North Atlantic (see Supplementary Fig. S9b). Although the cyclone does not strongly intensify, it facilitates the transport of warm and moist air masses from the south towards the river catchment. During two events over the Elbe catchment, a large upper-level cut-off low over southeastern Europe moves northward to Poland in combination with the intensification of a surface cyclone to the north and moisture transport also from the Black Sea region. Two other events are characterised by an omega blocking centered over Scandinavia and a large ridge over Central/Eastern Europe in between two lows over the eastern North Atlantic and Eastern Europe, but no clear surface cyclone (see Supplementary Fig. S9c for one case). Moist air masses are transported from Southern to Central Europe ahead of the western low pressure system, and lifting over the catchment is associated with a near-surface convergence zone. Additionally, an event over the Oder catchment is linked to a "high over low" blocking system over the British Isles and a convergence zone over the catchment (see Supplementary Fig. S9d). Three events over the Rhine catchment are characterised by a positive upper-level geopotential height anomaly over Eastern Europe with an upper-level trough that slowly moves from the British Isles towards the catchment while surface cyclones develop north of the Alps.

In addition to the primarily synoptic-scale environment and processes investigated in the composites of Fig. 5, also convective processes may influence the intensity of MEPEs. To quantify such a potential influence, we analyse composites of the convective available potential energy (CAPE) on the day of the events and the previous day (see Supplementary Fig. S10a for a CAPE composite at the day of Danube events). The composites show very low CAPE over Central Europe and just weak to medium CAPE over eastern Europe on both days (similar to all other catchments). For single events, there are some areas with weak CAPE in most of the cases, but only 1-3 events per catchment are associated with medium to high CAPE values over or nearby the specific catchment. This indicates that convective processes, here measured in terms of CAPE, appear to play a minor role for very extreme, 100-year precipitation events in the large river catchments investigated in this study. Note, however, that this conclusion may be sensitive to the underlying dataset, since the simulations applied here do not explicitly resolve moist convection.

### 4.2.2   Differences to 20-50-year precipitation events

The previous section has shown that the atmospheric conditions during 100-year precipitation events are often similar between the different river catchments. In this section, differences between 100-year and more moderate, 20-50-year precipitation events are identified through differences between the MEPE and LEPE composites. Figures 7 and 8 show such composite differences for the Danube and Elbe catchments, respectively, as different mechanisms distinguish MEPEs over these catchments.

LEPEs over the Danube catchment are associated with a similar upper-level pattern as MEPEs, with an upper-level cut-off low located over Central Europe centered slightly south of the river catchment in combination with ridges over the North Atlantic and Eastern Europe (Fig. 7a). Also the surface pressure configuration is similar with a large area of low pressure over Eastern Europe and high pressure over the North Atlantic (Fig. 7b). However, MEPEs are characterised by a significantly intensified low pressure anomaly at 500 hPa that also extends further south compared to LEPEs. In addition, the surface pressure is significantly lower over large parts of Central and Eastern Europe. Pressure is also lower around Iceland both at the surface and in the middle troposphere. The intensified cut-off low and associated cyclonic circulation during MEPEs is associated with intensified cold air advection, bringing colder air masses over the Mediterranean Sea along the southern flank of the cut-off low (Fig. 7c). Over the Danube catchment itself, no significant temperature difference occurs, but still the temperature is slightly higher during MEPEs northeast of the catchment. There is also no particular increase of the moisture content in the lower troposphere over most parts of Central and Eastern Europe, although some areas are characterised by minor increases during MEPEs (Fig. 7d). Figures 7a and b thus clearly show that dynamical mechanisms such as an intensified cut-off low at 500 hPa and intensified low pressure systems at the surface play a more important role than thermodynamic mechanisms (larger temperature or moisture content in the lower troposphere) in distinguishing 100-year extreme precipitation events from more moderate extreme events. Similar results are found for the Oder catchment (see Supplementary Fig. S12), where, in addition, also the 500 hPa ridge over the North Atlantic is slightly intensified during MEPEs. Furthermore, the lower-tropospheric moisture content is significantly reduced during MEPEs over several areas of Central Europe in combination with reduced lower-tropospheric temperatures.

Not in all catchments are MEPEs characterised by intensified circulation anomalies in comparison to LEPEs. In order to demonstrate different intensification mechanisms, composite differences between MEPEs and LEPEs are shown in Fig. 8 for the Elbe catchment. MEPEs over this catchment are rather associated with a slight northeastward shift of the 500 hPa low towards the east of the catchment instead of a clear intensification (Fig. 8a). Over the North Atlantic, the height of the 500 hPa surface is lower over Iceland and the ridge west of Ireland is slightly intensified during MEPEs. Similar results are found for sea level pressure (Fig. 8b). The local sea level pressure minimum over Eastern Europe is shifted northeastwards during MEPEs rather than being intensified compared to LEPEs. On a larger scale, sea level pressure values are generally lower over the northern part of the domain, but higher in the southwest during MEPEs. On the contrary, as opposed to the Danube catchment, the lower-tropospheric temperature over large parts of Eastern Europe is higher during MEPEs compared to LEPEs over the Elbe catchment, even though this difference is not statistically significant (Fig. 8c). The local cold anomaly over the Czech Republic and southern Poland is caused by the extreme precipitation event (e.g., due to cloud radiative cooling) and is not visible in the hours before the event (not shown). These generally higher temperatures go along with significantly increased lower-tropospheric moisture content, in particular in the region northeast of the sea level pressure minimum (Fig. 8d). This moisture can then be transported towards the Elbe catchment by the cyclonic circulation. In summary, more favourable thermodynamic conditions, in particular a higher atmospheric moisture content, thus distinguish MEPEs over the Elbe catchment. The circulation anomalies associated with these events are spatially shifted rather than intensified compared to LEPEs. Similar results are obtained for the Rhine catchment (see Supplementary Fig. S13). MEPEs over the Rhine catchment are characterised

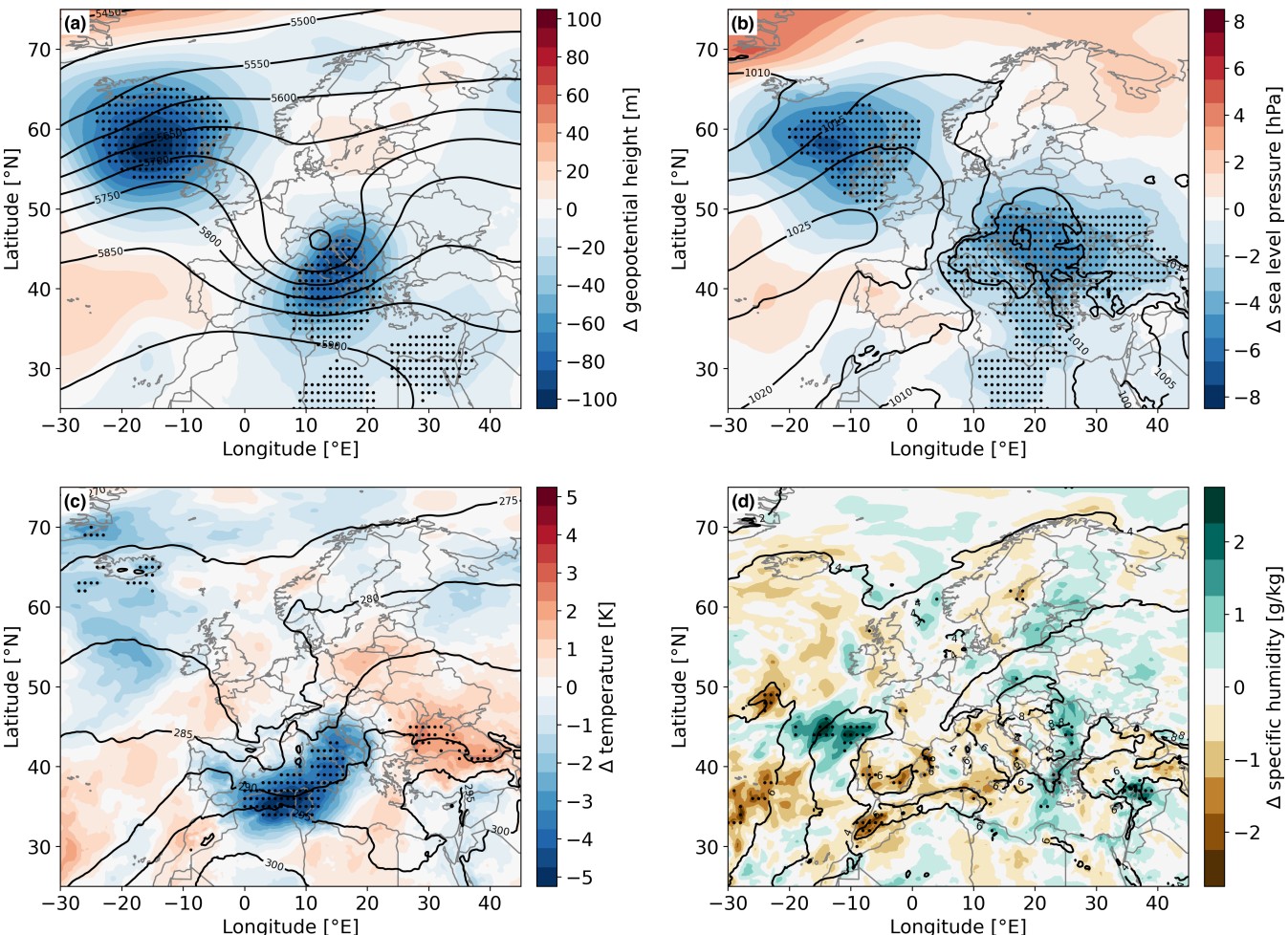

**Figure 7.** Differences between the composites of MEPEs and LEPEs 12 hours after the start of daily MEPEs in the Danube catchment, showing **(a)** geopotential height at 500 hPa, **(b)** sea level pressure, **(c)** temperature at 850 hPa as well as **(d)** specific humidity at 850 hPa. Positive (negative) values indicate higher (lower) values in the MEPE composite. Significant differences are marked by black dots. The LEPE composites are shown as black contours.

by a southward shift of the mid-level and surface low pressure anomalies in combination with a spatially large and significant intensification of an upper-level ridge over Northern and Eastern Europe. Additionally, temperature and lower-tropospheric moisture content are significantly higher during MEPEs over most of Eastern Europe. In the Weser/Ems catchment, both dynamic and thermodynamic mechanisms play a role for the differences between MEPEs and LEPEs (see Supplementary Fig. S14). The negative 500 hPa geopotential height anomaly is slightly enhanced and the surface low is significantly intensified during MEPEs. Both temperature and lower-tropospheric moisture content are significantly higher over parts of Eastern Europe, but conditions are even drier over large parts of Central and Western Europe.

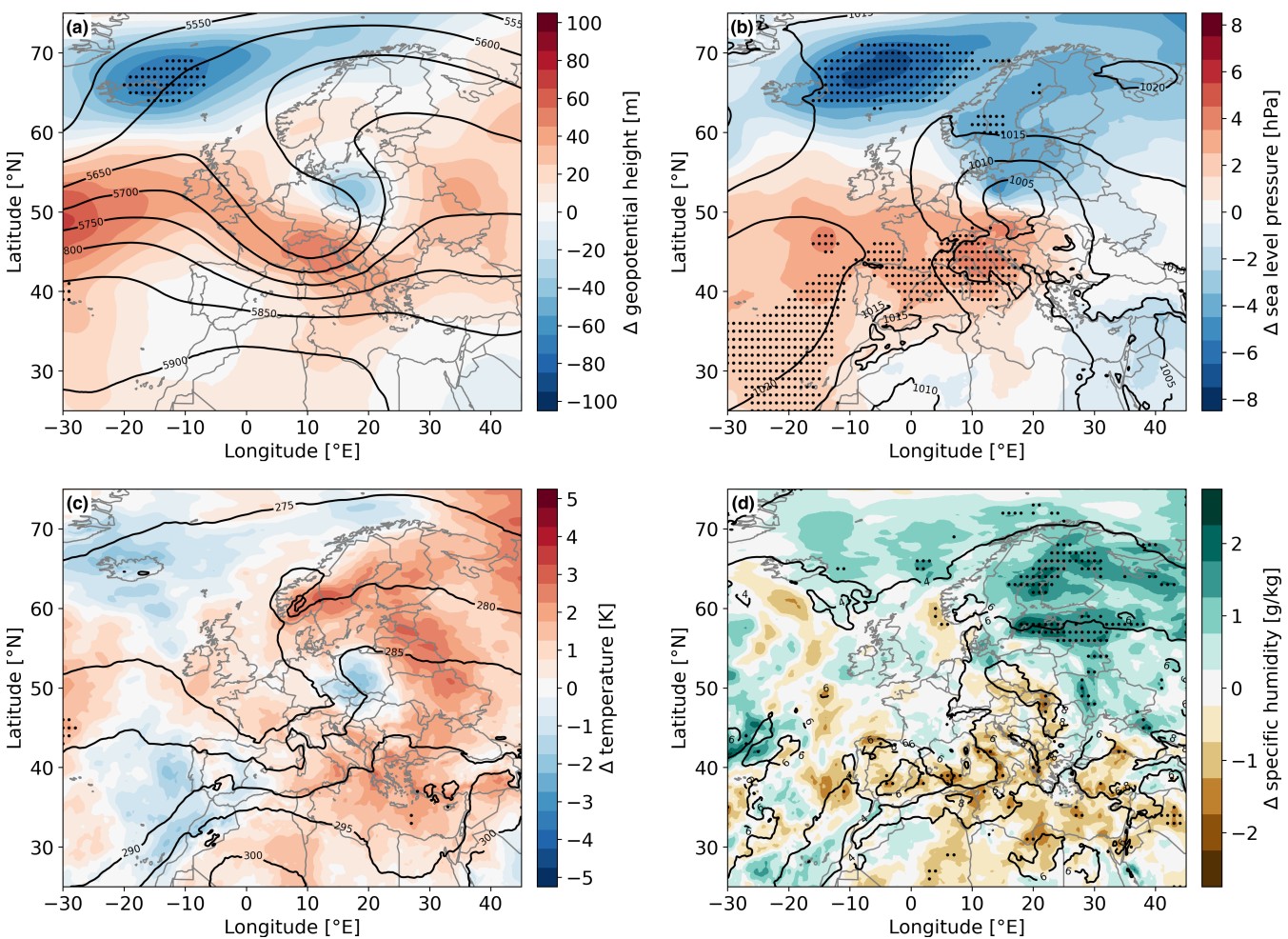

**Figure 8.** Same as in Fig. 7, but for events over the Elbe catchment.

Difference in geopotential height at 500 hPa as shown in Fig. 7a and Fig. 8a are an indicator of differences in the intensity of
the upper-level low intensity as well as the general synoptic situation. However, differences in the wind that, for instance, trans-
ports moist air masses towards the catchment, are more directly linked to the horizontal geopotential gradient. Supplementary
Fig. S11 shows the MEPE composites (left panels) and the differences to the LEPE composites (right panels) of the magnitude
of the geopotential height gradient at 500 hPa for the Danube (upper panels) and Elbe (lower panels) catchments. The largest
horizontal geopotential height gradients during MEPEs are found south/southeast of the centre of the upper-level cut-off low.
Local maxima are also found north/northeast of its centre, near the respective river catchment (this applies to all catchments).
In comparison with more moderate extreme events, MEPEs show (mostly) significant intensified horizontal gradients over the
eastern flank of the upper-level low centre. This is the case for all catchments, except for the Rhine (not shown). However,
the intensification of the horizontal gradient extends more northward for events over the Elbe compared to the Danube catch-

ment. The results for the Danube support our previous conclusion that the upper-level low is primarily more intense during
MEPEs, but not dislocated, leading to an intensified horizontal gradient particularly at its southern flank. However, for the
Elbe catchment the upper-level low centre shifts towards Poland during MEPEs, which increases the horizontal geopotential
height gradient over Eastern Europe and thus intensifies the northward advection of warm and moist air masses. Most probably,
this is the main reason for the increased temperature and specific humidity over northeastern Europe for MEPEs in the Elbe
catchment (see Fig. 8c and 8d). For the Elbe catchment, the role of altered dynamical conditions during MEPEs is thus more
subtle: while the intensity of the upper-level cut-off low is not substantially increased, a shift of the circulation pattern leads to
increased southerly advection and thus more favourable thermodynamic conditions for extreme precipitation.

Finally, there is no significant difference in CAPE over Central Europe between MEPEs and LEPEs (see Supplementary
Fig. S10b for the Danube) in all river catchments except for the Rhine, where CAPE is even reduced during MEPEs in some
areas. This supports our assumption that convective instability is not a major discriminator of extreme precipitation in our large
catchment areas.

## 4.3 Cyclone development

The composite method to identify typical atmospheric patterns associated with extreme precipitation events is complemented
by an event-based approach to objectively identify and track surface cyclones as described in section 3.5. Cyclones are asso-
ciates with both MEPEs and LEPEs at the day of the event and tracks are connected backward in time until cyclone genesis.
The positions of the surface cyclone centers associated with MEPEs are shown as coloured symbols in Fig. 6, the cyclone
frequency during MEPEs (taking all grid points within the outermost closed sea level pressure contour surrounding a sea
level minimum into account) in Fig. 9a. Most cyclones emerge downstream of the developing 500 hPa cut-off low (see colour
shading in Fig. 6) on the day before or the day of the MEPEs. This area downstream of an upper-level positive potential vor-
ticity anomaly is generally favourable for cyclogenesis (e.g. Hoskins et al., 1985). Most surface cyclones develop south of
the Alps and move in parallel with the mid-tropospheric low along the Alps and towards the east or southeast of the Danube
catchment. Such cyclone pathways are represented by the "Vb" type in the categorisation of van Bebber (1891). The cyclone
frequency during MEPEs has a maximum southeast of the Danube catchment, with values around 80% over northern Croatia
and western Hungary (see again Fig. 9a). The cyclone tracking results for MEPEs over the Elbe and Oder catchments are sim-
ilar to the Danube catchment, with cyclones also developing south of the Alps, but also over Eastern Europe, and then moving
northwards. Surface cyclones during MEPEs over the Rhine catchment rather develop north of the Alps or over the western
Mediterranean and track northwards, while those associated with MEPEs over the Weser/Ems catchment mainly develop over
southern and eastern Germany. Cyclones associated with LEPEs typically follow similar pathways as described for MEPEs,
with only a few exceptions, in all river catchments (not shown). Nevertheless, relative cyclone frequencies during LEPEs are
smaller than for MEPEs (see Table 2 and, for the Danube catchment, Fig. 9b), meaning that a smaller percentage of LEPEs
events is linked to a tracked surface cyclone. Moreover, also the cyclone intensity, measured here simply by the averaged mini-
mum sea level pressure, is enhanced for MEPEs compared to LEPEs (see again Table 2). These intensity differences are larger
for the Weser/Ems, Oder and Danube catchments, and even statistically significant for the former two, compared to Rhine and

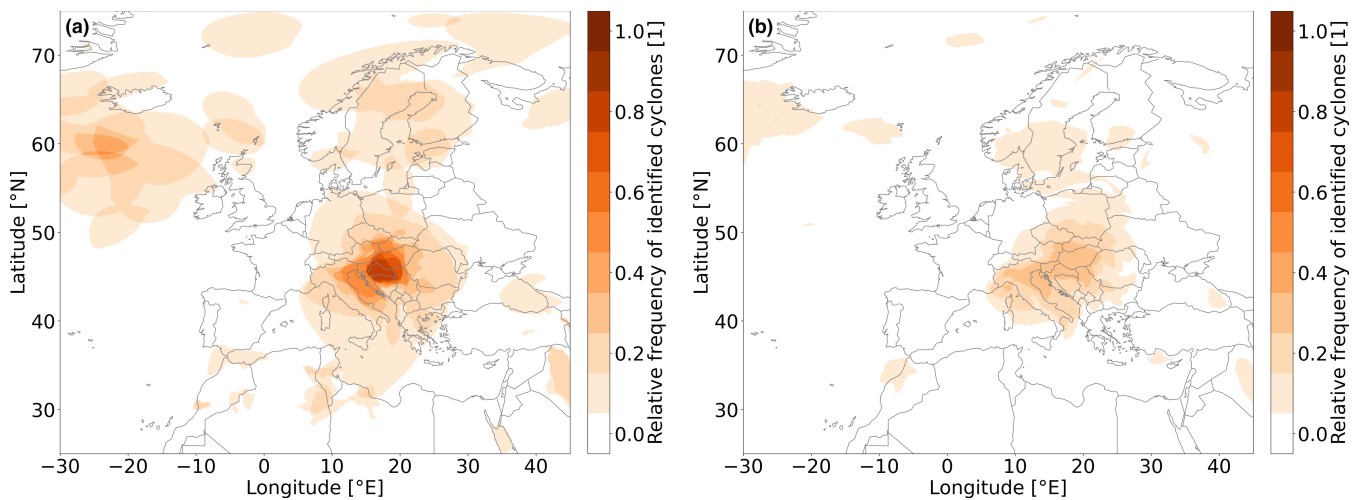

**Figure 9.** Cyclone frequency at 6 UTC on the day of **(a)** MEPEs and **(b)** LEPEs over the Danube catchment. All grid points within the outermost closed sea level pressure contour surrounding a sea level minimum are taken into account for the calculation of this frequency, following previous studies (Pfahl, 2014; Wernli and Schwierz, 2006).

| Catchments | Cyclone frequency | | | | Mean minimum SLP | |
|:---:|:---:|:---:|:---:|:---:|:---:|:---:|
| | MEPEs | | LEPEs | | MEPEs | LEPEs |
| Rhine | 11/13 | (84.6%) | 27/42 | (64.3%) | 999.0 hPa | 1002.5 hPa |
| Weser/Ems | 10/10 | (100.0%) | 20/29 | (69.0%) | **992.4 hPa** | **999.6 hPa** |
| Elbe | 10/11 | (90.9%) | 30/38 | (79.0%) | 995.3 hPa | 996.6 hPa |
| Oder | 12/13 | (92.3%) | 24/36 | (66.7%) | **992.8 hPa** | **999.7 hPa** |
| Danube | 11/13 | (84.6%) | 21/41 | (51.2%) | 995.8 hPa | 999.5 hPa |

**Table 2.** Absolute and relative cyclone frequencies as well as mean cyclone intensities, measured through the minimum sea level pressure, of cyclones associated with MEPEs and LEPEs. Bold SLP indicate significant differences on a significance level of 0.05.

Elbe. This is consistent with the result from the composite analysis that dynamical factors distinguish MEPEs from LEPEs in the former three catchments (see section 4.2.2).

In addition to the cyclone pathway and intensity, its velocity may be an important factor for extreme precipitation, as slower moving systems can lead to larger precipitation accumulation at a specific location. In order to analyse cyclone velocities, Fig. 10 shows the travelled distances of tracked cyclones in 6-hourly intervals for MEPEs (in red) and LEPEs (in blue). Only on the day of the events and the previous day, reasonable distributions can be produced as the number of identified cyclones is too low during earlier periods (see grey bars in Fig. 10). In general, cyclones tend to decelerate on the day of the event compared to

the previous day, which is consistent with the hypothesis that slower moving cyclones can enhance the precipitation amounts in the river catchments. However, comparing cyclone velocities between MEPEs and LEPEs during the day of the extreme

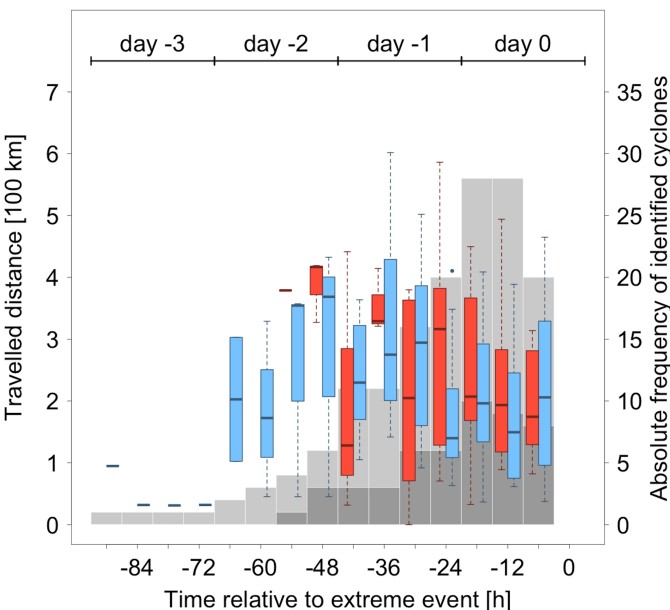

**Figure 10.** Distributions of travelled distances during 6-hourly time intervals of tracked cyclones during (red) MEPEs and (blue) LEPEs over the Danube catchment. The grey bars show the absolute cyclone counts at each time step for (dark grey) MEPEs and (light grey) LEPEs. The time on the x-axis is specified relative to the occurrence of the extreme event, and 0 hours corresponds to the end of the daily event.

events shows that there is a large scatter and no clear difference of the median velocities. Similar results are found for the other river catchments (not shown). Only for the Elbe and Oder catchments, there is a slight tendency towards slower cyclones for MEPEs compared to LEPEs.

In summary, the results of the cyclone tracking analysis show that extreme precipitation events in the selected river catchments are associated with typical cyclone tracks from the south towards the east of the respective catchment, and with a deceleration of the cyclones during the day of the event. MEPEs are linked to higher cyclone frequencies and also more intense cyclones compared to LEPEs, in particular in the Oder, Danube and Weser/Ems catchments. However, the cyclones' translation velocities are similar for MEPEs and LEPEs and thus do not allow distinguishing 100-year from less extreme precipitation

events.

## 5   Discussion and conclusion

The aim of this study has been to robustly investigate the atmospheric processes during 100-year precipitation events in large Central European river catchments and their differences compared to less extreme events. A better understanding of these high-impact events may be instrumental to provide more accurate forecasts of river floods and reduce uncertainties in projections

of their potential future changes. To perform such a robust process analysis, we use a large dataset of model-generated daily precipitation fields obtained from the ECMWF operational ensemble weather prediction system following a method introduced

by Breivik et al. (2013). Statistical analyses show that the 10th forecast days from these simulations produce quasi-independent precipitation events, in particular with regard to extremes such as annual maxima, and that the intensity distribution of these simulated events lies within the uncertainties obtained from different observational datasets. With the help of extreme value

theory, return levels of 100-year and more moderate, 20-50-year extreme precipitation events are determined for each selected river catchment, and the events surpassing these return levels are investigated with regard to the underlying atmospheric conditions.

An analysis of the temporal and spatial characteristics of the extreme precipitation events shows that most 100-year events occur during the extended summer June-September, with only a few exceptions in May or October. This corresponds well

with historical precipitation extremes (e.g., Grams et al., 2014; Ulbrich et al., 2003) and statistical analyses of rain gauge data (Fischer et al., 2016). More moderate precipitation extremes also peak during the summer season, but may occur during winter in the Rhine and Weser/Ems catchments as well. The spatial distribution of precipitation during the extreme events is influenced by the topography of the catchments, in particular by the orographic effect of the Alps in the Danube and Rhine catchments.

In order to determine the characteristic atmospheric conditions associated with the 100-year precipitation events, a composite

analysis is used in combination with a cyclone tracking algorithm. On the day before the events, an upper-level trough is typically located over Western Europe, which slowly moves southeastwards in the direction of the Alps. Rossby wave breaking (cf. Portmann et al., 2021; Appenzeller et al., 1996) then leads to the formation of a cut-off low in the middle and upper troposphere that is visible as both a geopotential height and potential vorticity anomaly. Previous studies also emphasised the importance of Rossby wave breaking for European precipitation extremes, in particular on the south side of the Alps (Barton et al., 2016;

Martius et al., 2008). The resulting cut-off low can favour the formation of extreme precipitation through its influence on moisture transport, lower-tropospheric destabilisation and dynamical lifting (Schlemmer et al., 2010). The exact pathway and location of the forming cut-off, typically towards the southeast of the specific river catchment, determines where the heavy precipitation event occurs. During the day of the event, the upper-level cut-off low becomes quasi-stationary, also associated with the formation of high pressure anomalies upstream over the North Atlantic and downstream over Eastern Europe. Surface

cyclogenesis regularly occurs east of the mid-tropospheric cut-off shortly before or during the extreme precipitation event. Cyclone pathways often, but not in all cases and for all catchments, resemble the classical "Vb" track (van Bebber, 1891), which has been associated also with many historical events (Hofstätter et al., 2018; Messmer et al., 2015; Grams et al., 2014; Bissolli et al., 2011; Mudelsee et al., 2004; Ulbrich et al., 2003). The involved cyclones are not necessarily very intense, consistent with previous findings (Pfahl and Wernli, 2012), but, in accordance with the upper-level anomalies, move slowly and thus lead

to high precipitation accumulation in the catchment. The spatial configuration with a cyclone typically located to the southeast of the affected catchment is again consistent with similar configurations found for less extreme precipitation events (Pfahl, 2014). It leads to moisture transport from the south around the cyclone centers and then towards the catchment with northerly or northeasterly winds. This is especially relevant for the Danube and Rhine catchments as a northerly flow towards the Alps enforces orographic precipitation enhancement. Previous case studies have shown that, during such situations, typically vari-

ous moisture sources contribute to the precipitation, with evapotranspiration from the land surface often playing the dominant role (Krug et al., 2022; Grams et al., 2014; Winschall et al., 2014). The important role of moisture transport in particular for

orographic precipitation associated with European floods has already been recognised in previous studies (Gvoždíková and Müller, 2021; Froidevaux and Martius, 2016).

To identify the mechanism that distinguish the very extreme, 100-year precipitation events (MEPEs), they are compared to a sample of less extreme events with return periods between 20 and 50 years (LEPEs). The general synoptic-scale patterns associated with LEPEs are similar to MEPEs, with an upper-level trough over Western Europe on the day before the event that often develops into a cut-off low, going along with surface cyclogenesis slightly downstream. The specific differences between MEPEs and LEPEs depend on the river catchment. On the one hand, differences in dynamical processes are most important for the Danube and Oder catchments. In particular, the cut-off low is intensified, surface cyclogenesis occurs more regularly and the surface cyclones are also more intense during MEPEs compared to LEPEs. This is expected to enhance moisture transport towards and dynamical forcing for ascent in the catchments, resulting in stronger precipitation. These results are in slight contrast to the findings of Pfahl and Wernli (2012), who did not detect differences in the intensity of cyclones associated with extreme precipitation in Central Europe. This is likely due to the fact that they analysed less extreme events than those studied here. The difference in cyclone intensity thus appears to be a specific characteristic of very extreme, 100-year events. However, there are no clear differences between the translation velocities of cyclones associated with MEPEs and LEPEs. On the other hand, MEPEs in the Elbe and Rhine catchments differ from LEPEs mainly due to significantly higher temperature and lower-tropospheric moisture content over large areas east and northeast of the river catchments. No pronounced differences in the strength of cyclones and cut-off lows are found for these catchments. The enhanced precipitation during MEPEs can thus mainly be explained by higher moisture content in the air masses transported towards the catchments, which are associated with enhanced northward flow due to a shift in the upper-level circulation anomalies. Finally, in the Weser/Ems catchment both stronger circulation anomalies and higher moisture content distinguish MEPEs from LEPEs.

The approach used in this study and, in particular, the analysis of extreme events in ECMWF ensemble prediction data comes with a number of limitations. A first limitation is the limited time span of 12 years (2008-2019) from which the forecast data are taken. Due to this relatively short time span, there is also a limitation in the sampling of large-scale boundary conditions associated with (multi-)decadal variability of the climate system. For instance, not the full spectrum of variability of the El Niño-Southern Oscillation and the Atlantic Multidecadal Oscillation is sampled, and it might be possible that even more extreme or structurally different events occur under boundary conditions that are not represented in these 12 years. This should be analysed in future research based on coupled climate model ensemble data. Second, there might be a signature of anthropogenic forcing in the dataset, leading to temporal inhomogeneities. Nevertheless, our analysis of the temporal homogeneity of the extreme events does not provide an indication of a trend due to anthropogenic climate warming. This is consistent with other studies showing that local/regional trends in extreme precipitation only emerge over longer periods of time (e.g. Fischer et al., 2013). Third, although our statistical analysis has shown that the different ensemble members are rather independent on day ten of the forecasts, there may still be some interdependence of extreme events occurring in different ensemble members initialised at the same day or on consecutive days, e.g., with regard to the synoptic-scale circulation. This would reduce the effective sample size of our composite and statistical significance analysis. Furthermore, the effective sample size of the entire dataset may also be reduced if spatially extended extreme precipitation events lead to simultaneous MEPEs/LEPEs in

several catchments. In our dataset, however, the number of such (quasi-)simultaneous events in several ensemble members or catchments is relatively small.

In this study, the first systematic analysis is presented of very extreme 100-year large-scale precipitation events in Central
Europe that goes beyond case studies. As the observational record is too short, we have to rely on simulated events, but a comparison to observations has shown that precipitation intensities are realistically represented in the applied ensemble prediction model. In future research, the approach may be used also for multi-day events, which have a high potential to cause flooding as described in section 1, with the help of more recent ECMWF forecast data extending beyond lead times of 10 days. Additionally, the method may be applied to other regions, as the dataset is available for the entire globe.

*Data availability.* The operational IFS ensemble forecast data from ECMWF can be obtained from https://apps.ecmwf.int/archive-catalogue/ ?type=cf&class=od&stream=enfo&expver=1 and https://apps.ecmwf.int/archive-catalogue/?type=pf&class=od&stream=enfo&expver=1 (last access: 25th April 2022). The user's affiliation needs to belong to an ECMWF member state. The observational dataset REGEN is freely available from https://geonetwork.nci.org.au/geonetwork/srv/eng/catalog.search#/metadata/f8555_9260_4736_9502 (last access: 25th April 2022) (Contractor et al., 2020). The observational data HYRAS can be accessed from https://gdk.gdi-de.org/geonetwork/srv/api/records/de.
dwd.hydromet.hyras.daily.info.status (last access: 25th April 2022). The observational E-OBS data can be downloaded from the ECA&D project via https://www.ecad.eu/download/ensembles/download.php (last access: 25th April 2022).

*Author contributions.* FR performed the analysis, produced the figures and drafted the manuscript. Both authors designed the study, discussed results and edited the manuscript.

*Competing interests.* SP is a member of the editorial board of Weather and Climate Dynamics. The authors have no other competing interests
to declare.

*Acknowledgements.* This study has been funded by the German Ministry of Education and Research (Bundesministerium für Bildung und Forschung, BMBF) in its strategy "Research for Sustainability" (FONA) in the framework of the "ClimXtreme" program, sub-project A2 (MExRain, grant number 01LP1901C). The German Meteorological Service DWD and ECMWF are acknowledged for providing the operational IFS model data. This work used resources of the Deutsches Klimarechenzentrum (DKRZ) which is granted by its Scientific Steering
Committee (WLA) under project ID bb1152. We are greatful to Heini Wernli (ETH Zurich) for his helpful comments on the event process analysis, to Felix Fauer and Henning Rust (both Freie Universität Berlin) for their helpful comments on the statistical analysis, to Hendrik Feldmann (Karlsruhe Institute of Technology) for the hint to possible observational datasets as well as to all colleagues of the "ClimXtreme" Module D for their technical assistance.

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
