# Peer review of "What distinguishes 100-year precipitation extremes over Central European river catchments from more moderate extreme events?"

_Weather and Climate Dynamics, 2022_

## Author Response (AR1)

**Responses to the comments of Reviewer 1 and 2**

by Florian Ruff and Stephan Pfahl

We would like to thank both reviewers for their helpful comments and suggestions to improve the manuscript! This document addresses all comments of reviewer 1 and 2 in the sections 1 and 2, respectively. The reviewer's comments are labelled in black and italic with the corresponding abbreviation RC1 and RC2, respectively. Our responses are given in blue with the abbreviation AC (author comments). In addition to modifications based on the reviewers' comments, we have added a small paragraph to the introduction section of the revised manuscript in which a relevant paper by Kelder et al. is discussed that we have not been aware of during the initial submission.

**1 Comments of Reviewer 1**

RC1: *Review of Ruff and Pfahl (2022): What distinguishes 100-year precipitation extremes over Central European river catchments from moderate extreme events.*

*In this paper, the authors study very extreme precipitation events over Central Europe river catchments, and the associated atmospheric dynamical conditions, in a robust way, thanks to the use of a very large ensemble of operational weather predictions. This clever and original approach allows them to study a much larger sample of extreme events than it would be possible with standard approaches based on observations or reanalyses.*

*The paper is generally well written, and discusses interesting results. But there are also some potential methodological issues, the methodological choices are not always well justified, and the implications of these choices are not always discussed. Therefore, I think that major revisions are needed.*

AC: Thank you very much for your helpful comments! We explain our methodological choices in more detail in the revised manuscript and discuss their implications and limitations in the discussion section.

**1.1 Reviewer 1: General comments**

RC1: *The authors could have done the same analyses with climate models outputs: large Single Model Initial-condition Large Ensembles (SMILEs) exist and could be used to obtain large sample sizes to do robust analyses of extreme precipitation events. The implicit hypothesis of the authors is, I think, that a weather prediction system provides more accurate representation of extreme precipitation events than climate models. I think this implicit hypothesis should be stated, and discussed, using references.*

AC: The reviewer is correct in stating that SMILEs can be used in a similar manner to obtain large sample sizes for studying very extreme events. Indeed, in the project from which this study emerged we are using both data from ensemble numerical weather prediction (NWP) and SMILEs. Both data sets have advantages and disadvantages. NWP ensemble data is available with higher resolution and from a very well calibrated model, but may, on the contrary, suffer from temporal inhomogeneities and inter-dependences between ensemble members. We are thus not sure that the NWP data are always „better", but we think that it is a still understudied approach to use these data for the analysis of extreme events. We are thus exploring this approach in detail focusing on European precipitation extremes. This will then also be used as a basis for comparing with SMILEs data. We discuss this more explicitly in section 2.1 of the revised manuscript.

RC1: *Precipitation observations are not assimilated in weather prediction systems, I think (is it the case here? It should be discussed), and, in the end precipitation is strongly the result of the atmospheric model, especially at day 10. So, what is really the advantage of a weather predictions system compared to a climate model? Only resolution?*

*As discussed below, there are also a few important drawbacks to the approach they follow compared to using SMILEs, and therefore I think it is important to discuss these points.*

AC:     It is correct that surface precipitation observations are not assimilated in ECMWF ensemble predictions. We provide more details on the NWP model in the data description of the revised manuscript. As mentioned above, advantages of the NWP model are the higher spatial resolution and the calibration of the model obtained from extensive comparison with observations on a daily basis.

RC1:    The authors use a clever and original approach to obtain a very large sample of extreme precipitation events to study, thanks to the use of the results of a large ensemble of weather predictions. But there are some unacknowledged limitations. Even if the ensemble of weather predictions is large, it spans a very short time period for climatological studies (2008-2019) and therefore samples a small sample of sea surface temperature (SST) conditions, for example. It means that it does not sample correctly interannual and low-frequency climate variability e.g. ENSO variations, decadal variations in NAO, AMV etc. These modes of variability or others may impact precipitation extremes. For example, what if the link between atmospheric circulation and extreme precipitation events is different during el Nino and la Nina events? Also, the 2008-2019 period is strongly impacted by anthropogenic forcings. This issue, and how it may impact the results of the study, should be discussed.

AC:     Thank you very much for bringing these points up. The limited time span together with the potential influence of natural variability is indeed a possible error source. We have added a more detailed discussion section describing this limitation, and also the other issues mentioned below. With respect to anthropogenic forcing, we think that it can also be an advantage to analyse a relatively short period during which trends due to anthropogenic forcing do not strongly influence the studied extreme events (see also the additional analyses on temporal homogeneity described below).

RC1:    More generally (as noted in the specific comments), the authors should sometimes better describe their analyses, better explain why they do them, how they reach their conclusions based on these analyses, and discuss their limitations. A part in the conclusion section should be dedicated to the discussion of the limitations of the analyses.

AC:     We tried to take these comments into account when preparing the revised manuscript and specifically added a more detailed discussion of the limitations of our approach.

RC1:    The physical analysis of extreme precipitation events could have been more developed. For example, the cyclone tracking algorithm is only based on SLP, which might not allow to capture the potentially complex atmospheric circulations associated with extreme precipitation. Also, the authors do not look at atmospheric stability, convective precursors etc., which are likely to play a very important role regarding these events.

AC:     The goal of the cyclone tracking analysis is to identify synoptic-scale circulation anomalies associated with the extreme events. We think that the applied algorithm is adequate for this purpose, and it is also clearly state of the art, as further discussed in our replies to the specific comments below. With regard to convective parameters, we have analysed convective available potential energy (CAPE) on the day of the events and the day before. Figure 1 shows the MEPE composite and the

[Figure]

**Figure 1.** (left) Composite of CAPE for all MEPEs and (right) difference between the MEPE and LEPE composites, all for events over Danube, 12 hours after the start of the events. For the difference plot, black dots mark significant changes and black contours indicate the LEPE composite.

differences to the LEPE composite for the Danube catchment. The composite shows almost no CAPE over Central Europe and just weak to medium CAPE over the eastern Europe. For single events, there are some areas with weak CAPE in most of the cases, but only one to three events per catchment are associated with medium to high CAPE values over or nearby the specific catchment. Additionally, there is also no significant difference in CAPE over Central Europe between MEPEs and LEPEs. Together, this indicates that convective instability is not a major discriminator of extreme precipitation in the large catchment areas we are analysing. We comment on this in the revised manuscript.

**1.2 Reviewer 1: Specific comments**

RC1: *Section 2.1*

*Very little is said on the weather prediction system. There are almost no references on the model, assimilation system, on the skill of the prediction system etc. Please add some information and references.*

*I suppose that the model has evolved during the period studied by the authors, with also changes in the assimilation system and observation networks etc. Am I right? We really need to have information on the evolutions of the system during the period studied by the authors.*

*Given these evolutions, there could be potential issues with the temporal homogeneity in the dataset studied by the authors and the analysis in Fig S1 is far from sufficient to show that it is not the case. At the very least, a discussion is needed on this point.*

AC: We have included more information and references regarding the description of the model and the assimilation system. The model was updated a few times within our selected time period. A collection of all changes (from Cycle 32r3, implemented before 1st Jan 2008, to Cycle 46r1, last implemented cycle before 31th Dec 2019) can be found in ECMWF (2023a). The full documentation of all the individual IFS model cycles can be found in ECMWF (2023b). There is also a Newsletter on the latest features of the model (ECMWF, 2023c). We now briefly discuss the updates in the description of the model data.

RC1: *L116. The authors cite Breivik (2013) to support the hypothesis that precipitation on day 10 of forecasts is independent. But this was with a previous version of the weather prediction system, I suppose. The skill of weather prediction systems increases with time and maybe it is not the case anymore?*

*Also, what the authors say, i.e. that precipitation from the different members on day 10 is independent implies that there is no predictability of precipitation at 10 days. Is it true? This should be discussed. Is it consistent with what we know about the skill of the weather prediction system? And even if it is true for precipitation, I'm quite sure it is not true for atmospheric circulation and that there is skill at day 10 for SLP, geopotential etc. So maybe precipitation itself from the different members is "independent "at day 10, but it is not the case for the associated atmospheric circulation, which is studied by the authors. What are the implications? I think it may be problematic, for example, to assess the equivalent sample size for the composites of large-scale circulation leading to extreme precipitation events. How is it done? In any cases, this general issue should be discussed.*

AC: Yes, the NWP system has been updated several times since the study of Breivik et al. (2013). Nonetheless, the correlations we find between the precipitation time series of different ensemble members are even lower than the correlations documented in Breivik et al. (2013) for wave height. This shows that the predictability on day ten indeed depends on the variable. Furthermore, the most crucial point for our analysis is not the independence of the entire precipitation time series, but rather the independent occurrence of extreme precipitation events in the different ensemble members, which is supported by the statistical analysis of annual maxima.

Correlations between the circulation of different ensemble members at day ten may be an issue if several MEPEs occur in different members for the same initialisation time. This only happens once in each of the Rhine, Danube and Oder catchments. This is only a very minor reduction of the equivalent sample size (actually by less than one, as the correlation between the circulation measures is not

[Figure]

**Figure 2.** Difference between the MEPE and LEPE composites of geopotential height at 500 hPa for the Danube catchment, 12 hours after the start of the events, using (a) all identified events (same as Fig. 7a in the original manuscript) and (b) just one event at dates with multiple events in different ensemble members. Black dots indicate statistical significance and black contours represent the LEPE composite.

large either). Also for the LEPEs, in spite of their larger number, it happens only three times that more than one event occurs at the same day for the Danube catchment, and less in the other catchments. To demonstrate that such small reductions in equivalent sample size do not substantially affect our results, Fig. 2 shows a comparison between Fig. 7a of the original manuscript (left) and the same composite analysis but with different sample sizes for MEPEs and LEPEs (right). The adjusted sample sizes (in Fig. 2b) just contain one event at dates with multiple events in different ensemble members, thus excluding 1 MEPE and 4 LEPEs. There are small differences between the composites on local scales, however, both the overall synoptic pattern for LEPEs (black contours) and the difference to MEPEs (colour shading), including the statistical significance, in Fig. 2b are very similar to those of Fig. 2a. The influence of simultaneously occurring events in different ensemble members is thus small. We have added a general discussion describing potential issues associated with reduced equivalent sample sizes due to simultaneous or consecutive events as well as events that affect several catchments at the same time to the revised manuscript.

RC1: *Figure 2a and b, and near line 205.*

*It is not totally clear to me how exactly the correlations and auto-correlations are computed. Are they computed at each point and then averaged on the catchment, or is precipitation spatially averaged before computing the correlations and auto-correlations? Is the annual cycle removed before computing the correlations and auto-correlations?*

*Also, are the correlations and auto-correlations calculated on the complete precipitation series or on the series with only the 10th forecast day? Based on section 2.1, I assume that it is the second possibility, but it is not so clear in section 3.2. Also, for each day there are two values, corresponding to two initializations, right? How is taken into account in the calculation of daily correlations and auto-correlations?*

*By the way, are the auto-correlations significant?*

AC: Thank you, as you assumed correctly and as described in section 2.1, just the 10th forecast day of each simulation is used. All initialisations are treated separately, also the two initialisations of each day, for which always the last 24 hours are used. For each of these (24-hour-period) precipitation field, the precipitation is spatially averaged over all grid points within the respective catchment. After that, the correlations are computed. We have tried to make this clearer in the revised manuscript.

To evaluate the influence of the annual cycle, which ranges from 0.74 mm to 5.34 mm (depending on the catchment), we tried removing it from the time series before computing the correlations. The correlation distribution for daily events just slightly changes (the mean correlation decreases from 0.18 to 0.15) and the influence on correlations between the strongest events is even smaller. Therefore, we decided to not further manipulate the data and keep the annual cycle as is.

We did not check the significance of the auto-correlations, but, as outlined below, this analysis has anyway been removed in the revised manuscript.

RC1: *Around L223. The reasoning behind the analyses in Figure 2a and 2b is not clear, and how exactly these analyses are linked to the hypothesis that precipitation from different members on the 10th forecast day is independent is not very clear. There are no real conclusions regarding this hypothesis.*

*For example, the authors write "to put the correlation coefficients between the times series into context and also to evaluate the auto-correlation of precipitation time series obtained from one ensemble member" as only justification to the analyses in Figure 2b, with no explicit connection with the previous hypothesis. And in the end, they don't conclude on the implications of the results for the hypothesis they want to prove.*

AC: The intention of Fig. 2a (original manuscript) is to show the low correlations between ensemble members with respect to precipitation at day ten, in particular when considering extreme events such as annual maxima. This is an important statistical justification for our assumption that these events can be treated as independent.

We agree that there is no clear value of Fig. 2b (original manuscript) to support this hypothesis and had included it just for comparison. As this apparently rather leads to confusion than clarification, we have removed it in the revised manuscript.

RC1: *L252. "can be considered independent"*

*What are exactly the criteria to consider them as independent? Could you describe the exact reasoning?*

AC: Our arguments are that, as described in the section before, the correlations between precipitation time series from different ensemble members are small (mean correlation of 0.18), and (almost) no significant correlations are obtained between extreme precipitation events in different members (in this case annual maxima that are used as block maxima to obtain the GEV). The summary sentence in the manuscript has been slightly extended to make this clearer.

RC1: *L253. "the data is thus suitable for systematic analysis of very extreme, 100-year precipitation events". Even if we consider precipitation on the 10th forecast day from the different members as independent, all data still only come from a 12-year period, and only sample 12 years of SST variability. As said in general comments, this is really problematic.*

*Note also that the period studied is strongly impacted by climate changes.*

*These points should absolutely be discussed, and the potential impacts on the conclusions of the paper clearly stated.*

AC: As mentioned above, we discuss the limitations associated with the limited time period of 12 year and potential effects of natural variability in more detail in the revised manuscript. See our response to your comment on line 260 below for the aspect of anthropogenic climate change.

RC1: *L259. What is the algorithm used to fit the parameters of the GEV distribution? Maximum likelihood?*

AC: Yes, the maximum-likelihood approach is used to fit the parameters. This information has been added to the revised manuscript.

RC1: *L260. How do the authors deal with the non-stationarity due to climate change? Over the period of the interest climate trends are very likely to be strong, and therefore the simple GEV model used by the authors, which makes a stationarity assumption, is likely to be quite inaccurate. Some approaches to take into account climate trends with GEV statistical models exist. Why didn't the authors use such an approach? The limitations of their method and its implications should at least should be discussed.*

AC: It is not our goal to study trends in the occurrence of extreme precipitation events and the influence of climate change, as a 12-year period is too short for this. We did several additional statistical

analyses to show that no trend can be detected in the extreme precipitation events obtained from NWP data over this period and that the analysed extreme events are distributed randomly over the period. The temporal distribution of the events is also shown in Fig. 3 below. First, no significant trend (according to the Mann-Kendall test, 95% confidence level) exists for the percentiles shown in Fig. S1 (original manuscript) in all catchments, except for the 0.999th percentile in Weser/Ems and Elbe catchments. The same test also indicates no significant trend in the number of MEPEs and LEPEs per year over the 12 years (taking data from all catchments together to obtain a sufficient sample size). Based on the Kolmogorov-Smirnov-Test, again with a 95% confidence level, the temporal distribution of both MEPEs and LEPEs (again for all catchments together) over the 12 years cannot be distinguished from a Poisson distribution, which is the case for independent events with a constant mean rate. Finally, in the individual catchments the occurrences of MEPEs and LEPEs per year almost all lie within the expected 95% confidence interval from a Poisson distribution with a constant mean rate (for MEPEs there is only one outlier in the Rhine catchment and for LEPEs two outliers in the Weser/Ems and Rhine catchments, which is both below 2% given the total numbers of 60 MEPEs and 186 LEPEs). This information has been added to the revised manuscript.

Hence, there is no detectable signature of climate warming in the extreme precipitation data over this 12-year period. This is consistent with other studies showing that local/regional trends in extreme precipitation only emerge over longer periods of time (e.g., Fischer et al., 2013). Accordingly, we don't consider it necessary nor useful (due to larger uncertainties in parameter estimates) to use GEV distributions with temporally varying parameters. Finally, we assume that this would also not substantially affect our main results, which are based on more than 10 events larger than the 100-year return level and should thus be relatively robust with regard to small changes in the estimate of this return level.

RC1:  L270-271.

Are all the daily precipitation events greater than the 100-year return level really used for the composite analysis (as it could be understood from the text) or only the events corresponding to block-maxima are used? I.e. if two consecutive days, or days in the same week (or month or semester) are above the 100-year return level, are they all used in the composite analyses? It does not really impact the composites, but it impacts their statistical significance, as it impacts the effective sample size.

AC:  All daily precipitation events that exceed the 100-year return level are used for the composites. The block maxima are just used to fit the GEV distribution and do not play a role in the composite analysis. The number of events occurring on the same day is still small (see our response to your comment on line 116 above). The limitations with regard to events in the same extended period (e.g., same week), which may be an issue due to uneven sampling of similar larger-scale boundary conditions such as SST, is discussed in the extended discussion section of the revised manuscript that also contains a discussion on the limited 12-year period.

RC1:  L285. The cyclone tracking algorithm is based only on SLP, which is quite basic regarding this kind of algorithm. Is it not problematic to track potentially complex situations leading to extreme precipitation events with such an algorithm? Is SLP not too "smooth" to capture correctly the complex dynamics associated with extreme precipitation events? Could the authors discuss the limitations of such approach or cite studies that show that it is OK to use such tracking algorithm for this kind of events?

AC:  The goal of the cyclone tracking analysis is to identify synoptic-scale circulation anomalies associated with the extreme events, and not the complex mesoscale conditions associated with the events. This is why a relatively smooth field such as SLP is adequate for this purpose. For instance, other very well established cyclone tracking schemes (Hodges, 1995) based on relative vorticity even explicitly truncate this field (to T42 spectral resolution) to retain only synoptic-scale features (see, e.g., Priestley and Catto, 2022).

The cyclone tracking algorithm used here is clearly state of the art and has been included in several recent inter-comparison studies (e.g., Neu et al., 2013; Flaounas et al., 2023). It has also been widely used to study the relationship between cyclones and extreme precipitation (e.g., Pfahl and Wernli, 2012; Pfahl, 2014). This is noted more explicitly in the revised manuscript.

*RC1: Figure 5. It is necessary to add statistical significance in the figure with composites, to demonstrate that the days with extreme precipitation events are really different from the other days. Without significance testing, we don't really know whether the authors discuss real signals or just statistical noise.*

AC: At this point we disagree with the reviewer. From a physical/meteorological point of view, it is totally obvious that extreme precipitation events have to be associated with specific atmospheric circulation patterns. This is why a test with the null hypothesis that the composite patterns cannot be distinguished from the annual or seasonal mean circulation does not make much sense to us. Also from a statistical point of view, we have shown that MEPE composite patterns differ significantly from composites of very similar, but less extreme events. There is no reason to believe that the differences from climatology (which are by construction larger than the differences between MEPEs and LEPEs, and for which the sample size is also much larger) should be not significant. Furthermore, such a significance test would be a lot of effort, as it would require us to download the atmospheric fields, which we currently have only for the MEPEs/LEPES, for the entire 12-year period from the NWP data archive (with a relatively slow data access). We don't think it is worth this effort to perform a test with a rather naive null hypothesis that would very likely be rejected anyway. Finally, we would like to add that recent statistical literature has also questioned the nonreflective application of p-value based significance tests (e.g., Wasserstein and Lazar, 2016), which should specifically apply to situations in which the test does not add new physical insights.

*RC1: L392-393. How do the authors know that "most of the individual events develop in a similar way as shown in these composites"? Is this based on a sort of test or analysis (e.g. clustering?) or just by looking at all events? If the second option is right, is it really sufficient?*

AC: This is indeed based on a subjective, visual analysis of the individual events. We think that, in this case, it is sufficient because indeed many events evolve in a very similar manner, which makes, for instance, a clustering analysis difficult and not very useful. To demonstrate this exemplarily, we show more individual events in the supplement.

*RC1: L 392-407.*

*It is somewhat strange to spend a long paragraph describing these specific events without showing them. They could be shown in SI.*

AC: We have included a few examples of individual events in the supplement.

*RC1: Fig 461. "in an area that is favourable for cyclogenesis". Why? The authors could cite some papers.*

AC: This statement is based on the typical baroclinic life cycle in which surface cyclogenesis occurs downstream of an upper-level positive potential vorticity anomaly (e.g., Hoskins et al., 1985). A note has been added to the text to make this clearer.

*RC1: L514. The authors discuss "Rossby wave breaking" at several places in the paper, even in the conclusion, but show no analysis of Rossby wave breaking.*

AC: Rossby wave breaking is the generic process through which the cut-off lows form that we see in our analysis of extreme precipitation events (e.g., Appenzeller et al., 1996; Portmann et al., 2021). We have included these references in the revised manuscript.

*RC1: L545. How do the authors explain the differences with Pfahl and Wernli (2012)?*

AC: As noted in the manuscript, this is likely due to the fact that Pfahl and Wernli analysed less extreme events. This difference in cyclone intensity thus appears to be a specific characteristic of very extreme, 100-year events. We have emphasised this more in the revised manuscript.

**2  Comments of Reviewer 2**

*RC2: For five Central-European catchments, authors study the characteristics of extreme precipitation events, mainly from the viewpoint of their causal atmospheric conditions. The study is focused on extreme events with the return period of areal daily precipitation total over 100 years, which the authors compare with a set of events of lesser intensity. For this, they use an innovative approach based on the processing of outputs from the ECMWF ensemble system.*

*In general, I find the study very interesting, innovative and well-written. Thus, I recommend it for publication in WCD after revisions with respect to the following comments.*

AC: Thank you very much for your comments! We have taken them into account in the revised manuscript as detailed below.

**2.1  Reviewer 2: General comments**

*RC2: Since operational outputs from the ECMWF model are used, I am not sure of the homogeneity of the input data (i.e. whether there was no major change in the model settings during the considered decade). This question needs to be discussed in more detail than only presenting time series of extreme percentile values of daily precipitation totals (S1). It would be necessary to verify whether the extreme events analyzed in the study were randomly distributed within the entire set of model outputs.*

AC: There were several updates of the model within our selected time period. We briefly discuss these updates in the revised manuscript. We did several additional statistical analyses to show that, nevertheless, no trend can be detected in the extreme precipitation events obtained from NWP data over this period and that the analysed extreme events are distributed randomly over the period. The temporal distribution of the events is also shown in Fig. 3. First, no significant trend (according to the Mann-Kendall test, 95% confidence level) exists for the percentiles shown in Fig. S1 (original manuscript) in all catchments, except for the 0.999th percentile in Weser/Ems and Elbe catchments. The same test also indicates no significant trend in the number of MEPEs and LEPEs per year over the 12 year (taking data from all catchments together to obtain a sufficient sample size). Based on the Kolmogorov-Smirnov-Test, again with a 95% confidence level, the temporal distribution of both MEPEs and LEPEs (again for all catchments together) over the 12 years cannot be distinguished from a Poisson distribution, which is the case for independent events with a constant mean rate. Finally, in the individual catchments the occurrences of the MEPEs and LEPEs per year almost all lie within the expected 95% confidence interval from a Poisson distribution with a constant mean rate (for MEPEs there is only one outlier in the Rhine catchment and for LEPEs two outliers in the Weser/Ems and Rhine catchments, which is both below 2% given the total numbers of 60 MEPEs and 186 LEPEs). This information has been added to the revised manuscript.

[Figure]

**Figure 3.** Yearly distribution of all MEPEs and LEPEs over all catchments combined.

RC2: *On the scale of large Central-European catchments, extreme precipitation events producing large floods regularly last more than one day. Authors mention this fact in conclusions but it is not enough in my opinion. Because accumulation of precipitation is a crucial factor of river floods, it should be mentioned already in the introduction as well as in the discussion at least. It could also happen that two days of extreme precipitation follow one after another in the dataset (as it was e.g. in July 1997) – in such a case, atmospheric conditions are certainly very similar on both days and it could influence the results.*

AC: Thank you, we have included statements about the importance of precipitation accumulated over several days in the introduction and discussion sections. However, given the data set and the selection of the 10th forecast day at the earliest, it is not possible to look at longer lasting events for all MEPEs/LEPEs. There is only one case in our data set in which two MEPEs of the same catchment occur at two consecutive days (seven cases for the LEPEs). Although the initial conditions are likely to be similar in these cases, after ten forecast days the simulations typically diverge. Having several events with similar initial conditions reduces the equivalent sample size of our composites (see also our response to the second specific comment of reviewer 1), but due to the small number of consecutive (and simultaneous) events, the influence on our results is small.

We have added a general discussion to the revised manuscript describing potential issues associated with reduced equivalent sample sizes due to simultaneous or consecutive events as well as events that affect several catchments at the same time to the revised manuscript.

RC2: *Moreover, extreme events usually affect more than one of the studied catchments (as e.g. in August 2002) but there is no mention of it in the article. It could be actually interesting and useful to have a 5x5 table giving the frequency of such events, with additional comments of possible cases when even more than two catchments were hit.*

AC: We have added remarks on the possibility of floods in neighbouring catchments triggered by spatially extended extreme precipitation events in the introduction and discussion sections. In our dataset, simultaneous MEPEs in two catchments occur in two cases (that is, for two initialisation dates). For LEPEs there are 16 such cases, i.e., a larger number, and in two of those even three catchments are simultaneously affected.

RC2: *Finally, authors distinguish between MEPEs and LEPEs to present what makes the events really extreme; however, there are also events with return periods between 50 and 100 which are worth noting in my opinion. Do they exhibit any "transitional patterns" between MEPEs and LEPEs?*

AC: Yes, there are still events between the 50- and 100-year return values which we do not analyse. Our reasoning behind this approach is that we would like to compare the MEPEs to a clearly distinct distribution of less extreme events. Nevertheless, based on the generally very similar circulation anomalies associated with LEPEs and MEPEs, it is very likely that also the events in between occur with a similar synoptic evolution, albeit with slightly different magnitudes of the circulation and/or moisture anomalies, as it is the case for LEPEs. We have added a remark on this point to the data section.

**2.2  Reviewer 2: Specific comments**

RC2: *l. 17-20: The March/April 2006 flood was not due to an extreme precipitation event; it was a typical snow-melt flood, with the melting process accelerated by the rain-on-snow process.*

AC: Thank you, we have rewritten the sentence and excluded this flood event from the list.

RC2: *l. 21: The sentence seems to mean that each of the mentioned flood events had such a large impact. Is it truth? I guess that less people died at least in March/April 2006 because snow-melt floods use to be well predicted in general.*

AC: We have slightly rewritten the sentence as „Many of these events had devastating impacts such as…".

*RC2: l. 28: In my opinion, the situation described is for Donau in Passau, not Elbe in Dresden – please, check Merz et al. (2014) again.*

AC: Thank you for this remark. We have changed the sentence.

*RC2: l. 42: The surface cyclogenesis is due to the upper-level circulation, not vice versa. Thus, I suggest to change the beginning of the sentence as follows: "With such an upper-tropospheric configuration, … is often associated."*

AC: We have rewritten the sentence in order to prevent misunderstandings.

*RC2: l. 184: It should be mentioned that only a rather small part of the whole Danube catchment is considered.*

AC: Thank you, we have clarified that not the entire catchment is considered here.

*RC2: l. 315: It is not clear whether the presented values are maximum values reached during the maximum event in each catchment or mean values calculated from the MEPEs. I fully agree that the highest values in the Weser/Ems region are due to the rather small area. To compare the catchments, it would be fine to present also maximum pixel values – I guess that the ranking of the catchments would be very different in this case.*

AC: The presented values are spatially averaged values over the specific catchment. For example, the 44.1 mm of the Danube catchment is the spatial average of all grid points within the catchment of the accumulated precipitation in Fig. 3a of the original manuscript. Higher values, of course, occur at single grid points. For the maximum grid-point values, the ranking changes as follows: Elbe (77.7 mm), Danube (74.3 mm), Rhine (68.1 mm), Weser/Ems (63.8 mm) and Oder (62.7 mm). We have added this information to the revised manuscript.

*RC2: Fig. 2a: I guess that the non-zero correlations between the entire daily time series of ensemble members could be due to the seasonal distribution of daily precipitation totals. Thus, the correlation analysis for two main seasons (DJF, JJA) would probably prove the independence of data better than the presented one.*

AC: We computed the correlation distributions for the two main seasons (JJA and DJF) and show the results in Fig. 4. The correlation is slightly lower for summer than for winter. Summer events are, thus, slightly more independent which is an important point as we mainly analyse summer extreme events. However, the differences between the seasons are rather small, and correlation coefficients for winter are even higher than for the entire year, indicating that the seasonality does not explain the slightly positive annual correlations.

[Figure]

**Figure 4.** Same as Fig. 2a (in the original manuscript) with additional distributions of Spearman correlation coefficients between daily summer/JJA (red box plot) and winter/DJF (blue box plot) precipitation time series.

*RC2: Fig. 2c: I do not fully understand why where is only one simulated total over 40 mm in comparison with REGEN and E-OBS but at least four of them in case of HYRAS. I guess that the simulated precipitation is the same for all datasets, so there should always be three marks at the same height in the graphs here and in S2.*

AC: The simulated precipitation for the quantile-quantile-plots (qq-plots) with REGEN and E-OBS observations are the same and they show the same values at the y-axis. However, we adjusted the simulated precipitation for plots with the HYRAS observations as this data set does not cover the entire catchment. In order to represent the quantile distributions of the same subregions of the catchment, we only use the simulated precipitation from grid points that are covered by at least 90% of HYRAS observations. For the Weser/Ems, the entire catchment is covered by HYRAS, and thus all three qq-plots show the same simulated precipitation (see supplementary Fig. S2b). We have added this explanation to the figure caption.

*RC2: Fig. 4: I find these results really interesting because extreme floods at Rhine, Weser and Ems usually appear in winter. Thus, it would be interesting to compare the monthly distribution of simulated extreme events with the monthly distribution of historical events, if there are data for this at least for some basins (it would be particularly interesting for the Rhine basin, where less extreme events extend into the cold part of the year).*

AC: Thank you for this comment. As the historical record is too short for a robust analysis of 100-year events, we have evaluated the seasonality of 10-year and 1-year precipitation events in the HYRAS data set, which has the highest spatial resolution. The results are shown in Fig. 5, which has been added to the supplement of the revised manuscript. Events over the Elbe, Oder and Danube (and partly Weser/Ems) have a clear maximum in summer, but weaker events sometimes occur in other months, which is generally consistent with our model-based analysis. However, the HYRAS observations show a different distribution for the Rhine catchment with events distributed rather evenly over the year and a slight maximum in the winter months. Given that these events are weaker than in our analysis, the seasonal distribution for the Rhine catchment might be shifting from winter to the summer season with increasing event intensity.

[Figure]

**Figure 5.** Monthly distribution of the (left) 7 and (right) 65 most intense precipitation events for each river catchment. The events can simply be considered as (left) 10- and (right) 1-year precipitation events.

*RC2: Fig. 8: I agree that the differences between MEPEs and LEPEs in 500 hPa geopotential height and SLP are less significant for Elbe, but what about the horizontal gradients of these variables? This is in my opinion much more important factor influencing the extremeness of subsequent precipitation.*

AC: Thank you very much for this comment, which prompted us to do additional analyses that led to more detailed insights into the dynamical differences between MEPEs and LEPEs. We did not have a look at the horizontal gradients of these variables before but we agree that they can be an important factor in order to better evaluate the dynamical situation. As the results for the SLP are quite noisy, Fig. 6 just shows MEPE composites (left panels) and differences to the LEPE composites (right) for

[Figure]

**Figure 6.** (left) MEPE composites of the absolute value of the geopotential height gradient at 500 hPa and (right) differences between MEPE and LEPE composites for events in the (upper row) Danube and (lower row) Elbe catchments, 12 hours after the start of the events. For the difference plots, black dots indicate statistical significance and black contours represent the LEPE composite.

the magnitude of the geopotential height gradient at 500 hPa. Upper panels show results for the Danube and lower panels for the Elbe catchment.

The largest horizontal geopotential height gradients during MEPEs are found south/southeast of the centre of the upper-level cut-off low. Local maxima are also found north/northeast of its centre, near the respective river catchment (this applies to all catchments). In comparison with more moderate extreme events, MEPEs show (mostly) significantly intensified horizontal gradients over the east flank of the upper-level low centre. This is found for all catchments except for the Rhine. However, the intensification of the horizontal gradient extends more northward for events over the Elbe compared to the Danube catchment. The results for the Danube support our previous conclusion that the upper-level low is primarily more intense during MEPEs, but not dislocated, leading to an intensified horizontal gradient particularly at its southern flank. However, for the Elbe catchment the upper-level low centre shifts towards Poland during MEPEs, which increases the horizontal geopotential height gradient over Eastern Europe and thus intensifies the northward advection of warm and moist air masses. Most probably, this is the main reason for the increased temperature and specific humidity over northeastern Europe documented in the original manuscript. For the Elbe catchment, the role of altered dynamical conditions during MEPEs is thus more subtle: while the intensity of the upper-level cut-off is not substantially increased, a shift of the circulation pattern leads to increased southerly advection and thus more favourable thermodynamic conditions for extreme precipitation. We have included this additional analysis and elaborated discussion in the revised manuscript.

**2.3 Reviewer 2: Technical comments**

*RC2: Terms "database" and "dataset" should be written as single words in my opinion.*

AC:     Changed.

RC2:    *If more than one MEPE or LEPE is considered, I recommend to use the abbreviations in the form MEPEs or LEPEs, respectively.*

AC:     Thanks, has been adapted.

RC2:    *l. 20: "when" should be written instead of "where";*

AC:     Corrected.

RC2:    *l. 285: "known" should be written instead of "know";*

AC:     Corrected.

RC2:    *l. 432: "catchment" should be written instead of "catchments";*

AC:     Corrected.

RC2:    *l. 479 and 481: "on the day" should be written instead of "at the day" in my opinion;*

AC:     Corrected.

RC2:    *Caption of Fig. 3: I suggest to mention already at the beginning of the caption that it is for all MEPEs but only in Danube catchment.*

AC:     Has been adapted.

RC2:    *Caption of Fig. 5: Because it is rather long, I would prefer if you start with a general sentence like "Composites of atmospheric conditions 12 hours after MEPEs started in the Danube catchment", followed by detailed description of individual maps.*

AC:     Thank you, we have rewritten the caption.

RC2:    *Fig. 6: The color scale should have the same intervals as in Fig. 5a.*

AC:     Has been adjusted.

RC2:    *Please, check titles of all referenced journals if they are correctly written (l. 656 – Monthly Weather Review; l. 631 – Quarterly Journal of the Royal Meteorological Society should be without the subtitle).*

AC:     Corrected.

RC2:    *Final comment: because the supplement is online only, it could be longer – readers would certainly appreciate graphs and maps for all considered catchments.*

AC:     Thank you, we have added more figures to the supplement.

**References**

Appenzeller, C., Davies, H., and Norton, W.: Fragmentation of stratospheric intrusions, Journal of Geophysical Research: Atmospheres, 101, 1435–1456, 1996.

Breivik, Ø., Aarnes, O. J., Bidlot, J.-R., Carrasco, A., and Saetra, Ø.: Wave extremes in the northeast Atlantic from ensemble forecasts, Journal of Climate, 26, 7525–7540, 2013.

ECMWF: Changes in ECMWF model, https://www.ecmwf.int/en/forecasts/documentation-and-support/changes-ecmwf-model, 2023a.

ECMWF: IFS documentation, https://www.ecmwf.int/en/publications/ifs-documentation, 2023b.

ECMWF: ECMWF Newsletter, https://www.ecmwf.int/en/publications/newsletters, 2023c.

Fischer, E. M., Beyerle, U., and Knutti, R.: Robust spatially aggregated projections of climate extremes, Nature Climate Change, 3, 1033–1038, 2013.

Flaounas, E., Aragão, L., Bernini, L., Dafis, S., Doiteau, B., Flocas, H., L Gray, S., Karwat, A., Kouroutzoglou, J., Lionello, P., et al.: A composite approach to produce reference datasets for extratropical cyclone tracks: Application to Mediterranean cyclones, Weather and Climate Dynamics Discussions, pp. 1– 32, 2023.

Hodges, K.: Feature tracking on the unit sphere, Monthly Weather Review, 123, 3458–3465, 1995.

Hoskins, B. J., McIntyre, M. E., and Robertson, A. W.: On the use and significance of isentropic potential vorticity maps, Quarterly Journal of the Royal Meteorological Society, 111, 877–946, 1985.

Neu, U., Akperov, M. G., Bellenbaum, N., Benestad, R., Blender, R., Caballero, R., Cocozza, A., Dacre, H. F., Feng, Y., Fraedrich, K., et al.: IMILAST: A community effort to intercompare extratropical cyclone detection and tracking algorithms, Bulletin of the American Meteorological Society, 94, 529–547, 2013.

Pfahl, S.: Characterising the relationship between weather extremes in Europe and synoptic circulation features, Natural Hazards and Earth System Sciences, 14, 1461–1475, 2014.

Pfahl, S. and Wernli, H.: Quantifying the relevance of cyclones for precipitation extremes, Journal of Climate, 25, 6770–6780, 2012.

Portmann, R., Sprenger, M., and Wernli, H.: The three-dimensional life cycles of potential vorticity cutoffs: A global and selected regional climatologies in ERA-Interim (1979–2018), Weather and Climate Dynamics, 2, 507–534, 2021.

Priestley, M. D. and Catto, J. L.: Improved representation of extratropical cyclone structure in HighResMIP models, Geophysical Research Letters, 49, e2021GL096 708, 2022.

Wasserstein, R. L. and Lazar, N. A.: The ASA statement on p-values: context, process, and purpose, 2016.